# The complex evolution and genomic dynamics of mating-type loci in *Cryptococcus* and *Kwoniella*

**Marco A. Coelho**[1ʘ*], **Márcia David-Palma**[1ʘ], **Seonju Marincowitz**[2], **Janneke Aylward**[2,3], **Nam Q. Pham**[2], **Andrey M. Yurkov**[4], **Brenda D. Wingfield**[2], **Michael J. Wingfield**[2], **Sheng Sun**[1], **Joseph Heitman**[1*]

**1** Department of Molecular Genetics and Microbiology, Duke University Medical Center, Durham, North Carolina, United States of America, **2** Department of Biochemistry, Genetics and Microbiology, Forestry and Agricultural Biotechnology Institute (FABI), University of Pretoria, Pretoria, South Africa, **3** Department of Conservation Ecology and Entomology, Stellenbosch University, Stellenbosch, South Africa, **4** Leibniz Institute DSMZ-German Collection of Microorganisms and Cell Cultures, Braunschweig, Germany

ʘ These authors contributed equally as first authors.
* marco.dias.coelho@duke.edu (MAC); heitm001@duke.edu (JH)

## Abstract

Sexual reproduction in basidiomycete fungi is governed by *MAT* loci (*P/R* and *HD*), which exhibit remarkable evolutionary plasticity, characterized by expansions, rearrangements, and gene losses often associated with mating system transitions. The sister genera *Cryptococcus* and *Kwoniella* provide a powerful framework for studying *MAT* loci evolution owing to their diverse reproductive strategies and distinct architectures, spanning bipolar and tetrapolar systems with either linked or unlinked *MAT* loci. Building on recent comparative genomic analyses, we generated additional chromosome-level assemblies, uncovering distinct trajectories shaping *MAT* loci organization. Contrasting with the small-scale expansions and gene acquisitions observed in *Kwoniella*, our analyses revealed independent expansions of the *P/R* locus in tetrapolar *Cryptococcus*, possibly driven by pheromone gene duplications. Notably, these expansions coincided with a pronounced GC-content reduction best explained by reduced GC-biased gene conversion following recombination suppression, rather than relaxed codon usage selection. Diverse modes of *MAT* locus linkage were also identified, including three previously unrecognized transitions: one resulting in a pseudobipolar arrangement and two leading to bipolarity. All three transitions involved translocations. In the pseudobipolar configuration, the *P/R* and *HD* loci remained on the same chromosome but genetically unlinked, whereas the bipolar transitions additionally featured rearrangements that fused the two loci into a nonrecombining region. Mating assays confirmed a sexual cycle in *Cryptococcus decagattii*, demonstrating its ability to undergo mating and sporulation. Progeny analysis in *Kwoniella mangrovensis* revealed substantial ploidy variation and aneuploidy, likely stemming from haploid–diploid mating, yet evidence of recombination and loss of heterozygosity indicates that meiotic exchange occurs despite irregular chromosome

**Data availability statement:** All primary data are within the paper and its Supporting information files. Genomic data was deposited in NCBI with BioProject accession numbers provided in S1 Appendix. The code developed for data analysis and figure generation are publicly available on Zenodo at https://doi.org/10.5281/zenodo.16987438.

**Funding:** This study was supported by the National Institute of Allergy and Infectious Diseases of the National Institutes of Health (https://www.niaid.nih.gov/) under awards R01 AI050113-20 (J.H.), R01 AI039115-27 (J.H.), and R01 AI33654-08 (J.H.). We note that these collaborative studies were concluded prior to June 27, 2025. Funding for the collection of Cryptococcus isolates from bark beetles in South Africa was supported entirely and independently by the Harry Oppenheimer Fellowship Award of The Oppenheimer Memorial Trust (https://www.omt.org.za/hfo-award) (to M.J.W.) and the SARChI Chair in Fungal Genomics, supported by the Department of Science and Innovation (https://www.dst.gov.za/) / National Research Foundation (https://www.nrf.ac.za/), South Africa (to B.D.W.). The funders had no role in study design, data collection and analysis, decision to publish, or preparation of the manuscript.

**Competing interests:** I have read the journal's policy and the authors of this manuscript have the following competing interests: J.H. serves on the Editorial Board of PLOS Biology and is a Co-Director and Fellow of the CIFAR program Fungal Kingdom: Threats & Opportunities (https://cifar.ca/research/program/fungal-kingdom/). All the other authors have declared that no competing interests exist.

**Abbreviations:** CAI, codon adaptation index; COGs, Cluster of Orthologous Genes; FACS, fluorescence-activated cell-sorting; gBGC, GC-biased gene conversion; gCF, gene concordance factor; LOH, loss of heterozygosity; LPP, local posterior probability; LTRs, long-terminal-repeat retrotransposons; ML, maximum likelihood; P/R, pheromone/receptor; RFLP, restriction fragment length polymorphism; sCF, site concordance factor; SC-OGs, single-copy orthologs; SH-aLRT, Shimodaira–Hasegawa approximate likelihood ratio test; TEs, transposable elements, UFboot, ultrafast bootstrap approximation.

segregation. Our findings underscore the importance of continued diversity sampling and provide further evidence for convergent evolution of fused *MAT* loci in basidiomycetes, offering new insights into the genetic and chromosomal changes driving reproductive transitions.

## Introduction

Sexual reproduction is a cornerstone of eukaryotic biology, generating genetic variation through recombination and allele segregation, which fuels adaptation in changing environments [1–3]. The Fungal Kingdom is amongst the most ecologically diverse, with species thriving in nearly every environment, acting as decomposers, symbionts, commensals, as well as pathogens of plants, animals, and other fungi [4–6]. This versatility is mirrored in a wide range of dispersal stages and reproductive strategies, from heterothallism, requiring genetically distinct individuals of opposite mating types, to homothallism, which allows self-fertilization by a single individual as well as outcrossing [7,8].

In heterothallic basidiomycete fungi, sexual compatibility is typically governed by two mating-type (*MAT*) loci. The pheromone/receptor (*P/R*) locus encodes at least one mating pheromone and one G-protein-coupled pheromone receptor, enabling mate recognition and fusion. The *HD* locus encodes two homeodomain transcription factors (HD1/Sxi1 and HD2/Sxi2) that heterodimerize post-mating to control dikaryotic growth and sexual development [9]. Successful mating occurs only between mating partners with different, compatible alleles of both *MAT* loci.

Most basidiomycetes exhibit a tetrapolar breeding system, with *P/R* and *HD* unlinked on separate chromosomes, allowing each meiotic event to generate up to four distinct mating types [10]. Some lineages, however, have evolved a bipolar system, in which sexual compatibility is controlled by a single *MAT* locus. This transition can occur either through loss of function of one locus in mating-type determination, as in *Coprinellus disseminatus* [11] and other mushroom-forming *Agaricomycetes* [12–14], or via physical linkage and subsequent fusion (i.e., genetic linkage) of the *P/R* and *HD* loci. In the latter case, translocations or chromosome fusions bring the two loci together, followed by additional rearrangements that suppress recombination, resulting in a single, fused, nonrecombining *MAT* region that is often expanded, highly rearranged, and biallelic [15–18]. Some basidiomycetes also exhibit a pseudo-bipolar configuration, where the two *MAT* loci lie on the same chromosome but remain genetically unlinked because they are sufficiently distant to allow recombination, as seen in skin-commensal/pathogenic *Malassezia* species [19,20]. Beyond heterothallism, several basidiomycetes reproduce without a partner. This strategy, broadly termed homothallism, encompasses diverse genetic mechanisms, including primary homothallism, where a single individual carries all *MAT* alleles necessary for sexual development [21–24], and unisexual reproduction involving cells of the same mating type [25].

PLOS Biology

Evidence suggests that tetrapolarity is ancestral in basidiomycetes, with bipolarity via *P/R-HD* fusion arising independently multiple times [15,16,18,26,27]. This bipolar configuration, often observed in fungal species associated with plants or animals as commensals or pathogens, increases sibling compatibility from 25% (tetrapolar) to 50% (bipolar), thereby facilitating inbreeding [9,28,29]. This may benefit species colonizing hosts, where spore dispersal and access to unrelated mating partners may be more limited. Bipolar species with fused *MAT* loci are found across the three basidiomycete subphyla, including smut fungi of grasses [15,30,31], anther-smut *Microbotryum* infecting *Caryophyllaceae* [17,27,32,33], and two *Tremellomycetes* lineages: *Trichosporonales* [18] and the human-pathogenic *Cryptococcus* species [16,34,35].

An important aspect of fused *MAT* loci is the inclusion of genes beyond the core *HD* and *P/R* determinants, some encoding regulators of post-mating development that influence morphogenesis (e.g., *STE20*, required for polarity maintenance in dikaryotic hyphae [36]) and dispersal (e.g., *SPO14*, likely involved in spore production) [37]. Additionally, recombination suppression often extends beyond the core mating-type genes, encompassing larger regions of the *MAT*-containing chromosome [38]. While this may preserve beneficial allele combinations, it also contributes to genomic degeneration, including gene loss, accumulation of transposable elements (TEs), and extensive rearrangements. These dynamics parallel sex chromosome evolution in other eukaryotes, highlighting potential convergent evolution in sexual reproduction regulation [34,39].

The genus *Cryptococcus* includes both human-pathogenic species of critical importance and nonpathogenic saprobes. The pathogenic clade comprises *Cryptococcus neoformans*, *C. deneoformans*, and six species in the *C. gattii* complex [40,41], with *C. neoformans* ranking first on the World Health Organization's list of critical fungal pathogens [42,43]. Nonpathogens include *C. wingfieldii*, *C. amylolentus*, *C. floricola*, *C. depauperatus*, and *C. luteus* [23,44–47], along with newly identified lineages [48,49], many associated with insects, particularly bark beetles [50]. The sister genus, *Kwoniella*, consists of saprophytes found in diverse environments, including soil, seawater, plant material, and insect frass [46,48,49,51,52].

Sexual reproduction is well characterized in *Cryptococcus* and has been linked to pathogenicity in medically relevant species [53–55]. A hallmark of the sexual cycle is the yeast-to-hypha transition, which occurs after mating between yeast cells of opposite mating types (**a** and α) or during unisexual reproduction [25]. This morphological switch aids nutrient foraging [56,57] and enhances survival against environmental stressors and microbial predators, thought to be major selective forces in *Cryptococcus* evolution [53,58–62]. Hyphae eventually differentiate into basidia, where diploid nuclei undergo meiosis to produce basidiospores that are smaller than yeast cells and more resilient to environmental stresses. These spores facilitate alveolar deposition upon inhalation and subsequent dissemination within the host [63–65]. In contrast, sexual reproduction in *Kwoniella* has been documented only in *K. mangrovensis* and *K. heveanensis* [52,66,67]. These species do not form aerial basidia with spore chains, likely making them less efficient at environmental dispersal than *Cryptococcus*.

Recent surveys have uncovered novel *Cryptococcus* and *Kwoniella* species [41,48,49,68,69], often from single isolates, limiting mating tests and hindering characterization of reproductive strategies and compatibility systems. In such cases, high-quality genome assemblies offer a powerful alternative, enabling comparative analyses of *MAT* loci to elucidate reproductive systems. While our recent work generated chromosome-level genome assemblies for several *Cryptococcus* and *Kwoniella* species, providing insights into their genomic architecture and evolution [49], this was restricted to single strains per species, precluding in-depth analyses of *MAT* locus variation.

Here, we examine a broader diversity of species and, where possible, strains of opposite mating types within the same species, enabling systematic characterization of *MAT* loci structure, chromosomal organization, and key evolutionary transitions. Our findings reveal tetrapolarity as the ancestral state of both *Cryptococcus* and *Kwoniella*, with *P/R-HD* loci fusion (i.e., genetic linkage) evolving independently three times: (i) in the common ancestor of pathogenic *Cryptococcus* species, (ii) in a recently identified nonpathogenic *Cryptococcus* species, and (iii) in one *Kwoniella* species. We further

show that the *P/R* locus itself is highly dynamic, sharing only a core set of four genes across both genera, and has undergone lineage-specific expansions, particularly in *Cryptococcus*, where increased pheromone gene copy number appears to have driven further rearrangements. Experimental crosses also confirmed a previously unobserved sexual cycle in *C. decagattii* and provided the first direct evidence of recombination in *K. mangrovensis*, revealing meiotic exchange in progeny and nonhaploid offspring with potential for self-fertility. Collectively, our findings support that fused *MAT* loci have evolved multiple times independently in basidiomycetes, reinforcing the role of convergent evolution in shaping reproductive strategies and ecological adaptation.

## Results

### Chromosome-level assemblies and updated phylogenetic relationships of *Cryptococcus* and *Kwoniella*

To enable detailed analyses of *MAT* locus structure and evolution, we expanded our previous genomic dataset (29 assemblies; [49]) by incorporating genome sequences for 12 additional *Cryptococcus* and 7 *Kwoniella* strains, including opposite mating types where available. Among the *Cryptococcus* strains, four represent three currently undescribed species (*Cryptococcus* sp. 3, sp. 4, and sp. 5; Fig 1A), while the remaining eight are opposite mating types of pathogenic *Cryptococcus* species previously sequenced [49]. For *Kwoniella*, three strains belong to the recently described species *K. ovata*, *K. endophytica*, and *K. fici* [48,70,71], while the remaining four strains are different mating types of *K. mangrovensis*, *K. europaea*, *K. botswanensis*, and *K. heveanensis*, as previously determined through PCR and mating tests [52,66,67]. All but three genomes were assembled from long-read sequencing data (PacBio or Oxford Nanopore) and polished with short-read (Illumina) data, yielding chromosome-level genome assemblies. The exceptions, *K. europaea* PYCC6162, *K. botswanensis* CBS12717, and *K. heveanensis* BCC8398 were sequenced solely with Illumina, resulting in more fragmented assemblies. Analyzed strains were inferred to be haploid based on genome sizes consistent with those previously documented for haploid strains [49]. In total, the expanded dataset comprises 27 *Cryptococcus* strains (spanning 17 species) and 22 *Kwoniella* strains (spanning 18 species) (S1 Appendix).

To establish evolutionary relationships across species and determine the placement of the newly sequenced taxa, we identified 3,086 single-copy orthologous genes shared across *Cryptococcus* and *Kwoniella*, along with three outgroup species (*Tremella mesenterica*, *Saitozyma podzolica*, and *Bullera alba*), and reconstructed phylogenies employing both Maximum Likelihood (ML) and coalescent-based methods. Both approaches produced largely congruent results, except for the variable placement of *Cryptococcus* clades B and C relative to clade A (Figs 1A and S1). This inconsistency, noted in earlier studies [23,45,47], remains unresolved with the current dataset. Despite this, phylogenetic placement of the newly identified *Cryptococcus* and *Kwoniella* species was resolved. *Cryptococcus* sp. 3, known from a single strain (CMW60451) isolated from a bark beetle (*Lanurgus* sp.) infesting twigs of the endangered conifer *Widdringtonia cedarbergensis* in the Cederberg Mountains of South Africa, was placed as the closest relative of *C. depauperatus*, albeit on a long branch. *Cryptococcus* sp. 4 [72] and *Cryptococcus* sp. 5 clustered with *Cryptococcus* sp. 6 (OR918) [49,73], together forming a sister clade to all other known *Cryptococcus*. Among *Kwoniella*, *K. ovata* and *K. endophytica* were identified as sister species closely related to *K. dendrophila*, while *K. fici* grouped with *Kwoniella* sp. 4. The new *Cryptococcus* and *Kwoniella* species will be formally described elsewhere.

### Sexual reproduction, breeding systems, and *MAT* gene identification and organization across *Cryptococcus* and *Kwoniella*

Sexual cycles have been described for eight *Cryptococcus* species: seven primarily heterothallic (*C. neoformans*, *C. deneoformans*, *C. gattii*, *C. bacillisporus*, *C. deuterogattii*, *C. amylolentus*, *C. floricola*), and one homothallic (*C. depauperatus*) (Fig 1A) [23,47,74–78]. In *Kwoniella*, sexual reproduction has been confirmed only in *K. mangrovensis* and *K. heveanensis*, both heterothallic [45,52,66]. We sought to identify sexual cycles in species where none had

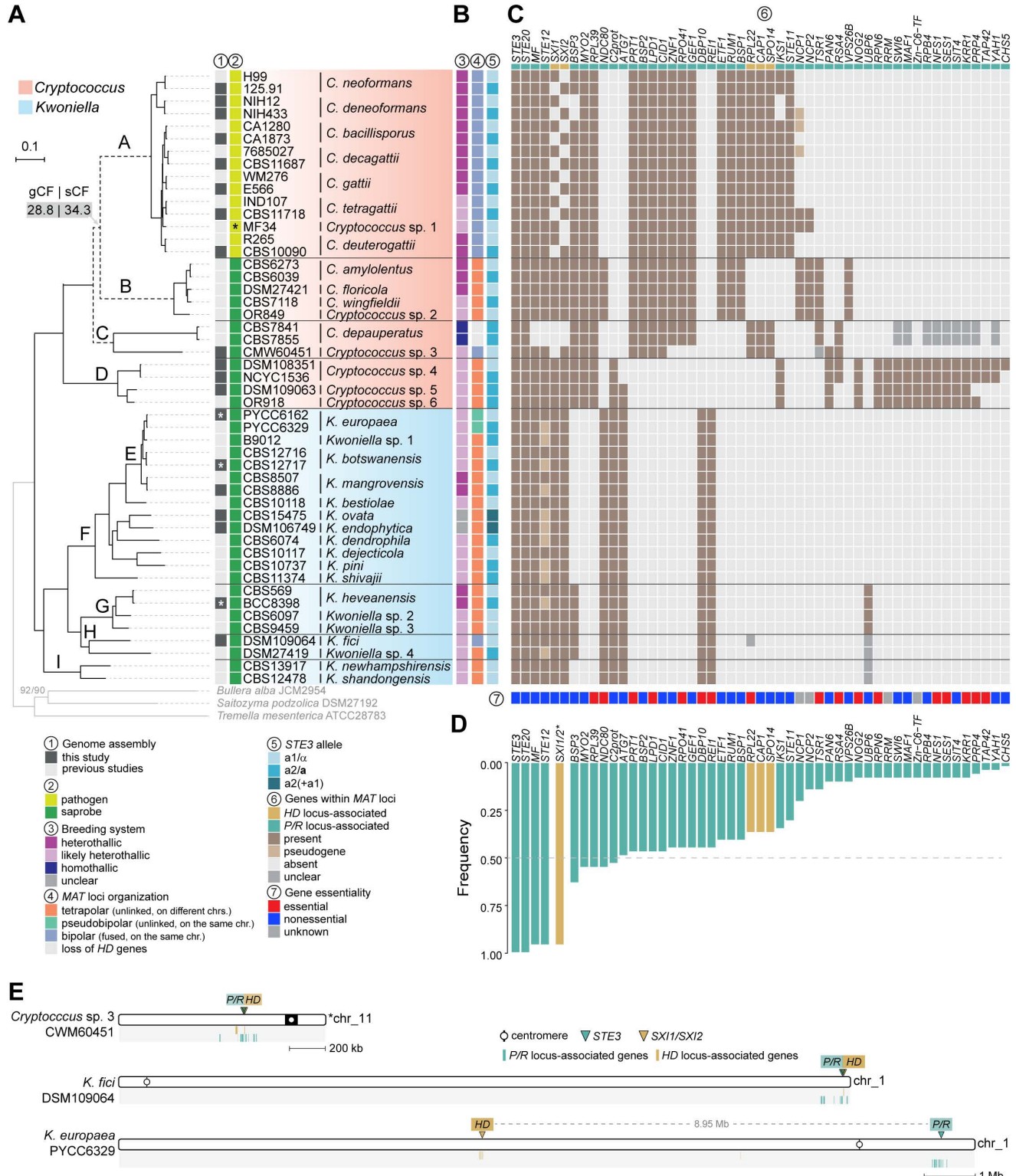

**Fig 1. Phylogeny, breeding systems, and *MAT* loci organization in *Cryptococcus* and *Kwoniella*. (A)** Maximum likelihood phylogeny of strains analyzed in this study, inferred from a concatenated data matrix comprising protein alignments of 3,086 single-copy genes. Clades are labeled as A–I, with clade E shown as a nested subclade within clade F for easier cross-referencing. Branch lengths are given as substitutions per site (scale bar). All

branches are supported >95% by SH-aLTR and UFBoot tests unless otherwise indicated. Gene concordance factor (gCF) and site concordance factor (sCF) were assessed to evaluate genealogical concordance, revealing unresolved branching order of clades A, B, and C (dashed branches; see S1 Fig for details). A black asterisk denotes that VGV has not yet been associated with human infections, and a white asterisk marks strains sequenced solely with Illumina, yielding more fragmented assemblies. **(B)** Predicted breeding systems, *MAT* locus organization, and mating-type identity of the *P/R* locus. Breeding systems are classified as heterothallic or homothallic for species with defined sexual cycles, and as likely heterothallic for species inferred to exhibit heterothallism based on *MAT* gene content. **(C)** Heatmap of gene presence/absence and pseudogene annotations within inferred *MAT* loci. Genes are ordered by their frequency of presence across species, with the most conserved genes on the left. Genes marked as "unclear" indicate cases where presence within *MAT* is uncertain because locus boundaries could not be precisely determined. *BSP3* and *IKS1*, evicted from the *MAT* locus in *C. deneoformans* [34], are listed as absent in this species. Gene essentiality, as predicted or experimentally validated in strain H99 in prior studies rather than direct assessments in each species, is displayed at the bottom. Genes scored as absent from *MAT* may still be present elsewhere in the genome. **(D)** Gene presence frequency plot summarizing panel **C.** Pseudogenes were treated as present for frequency calculations, while unclear cases were treated as absent. The identification of either *SXI1* or *SXI2* was scored as "present" and combined for frequency quantification. **(E)** *MAT* loci locations in *Cryptococcus* sp. 3 (CWM60451), *K. fici*, and *K. europaea*, illustrating bipolar and pseudobipolar arrangements, respectively. Genes typically found either within or near the *HD* and *P/R* loci in tetrapolar species are designated as *HD-* and *P/R*-associated genes, and their respective chromosomal locations are indicated by gold or teal vertical bars. The data underlying this figure can be found in S1 Appendix and at https://doi.org/10.5281/zenodo.16987438.

been described, restricting tests to cases where opposite mating types were available. Crosses were performed under conditions known to induce mating in *Cryptococcus* [79] and examined for the presence of hyphal growth or other sexual structures (basidia and basidiospores).

In *C. decagattii*, sexual reproduction was confirmed by crossing strains 7685027 (*MAT*α) and CBS11687 (*MAT***a**), which produced sexual structures (S2A Fig). In contrast, crosses between *C. tetragattii* IND107 and CBS11718 did not yield sexual structures (S2B Fig). For *Cryptococcus* sp. 3, sister to *C. depauperatus*, we observed key differences: while *C. depauperatus* is homothallic and exhibits continuous hyphal growth [23], *Cryptococcus* sp. 3 grew mostly as yeast and did not produce sexual structures alone, suggesting it is unlikely to be homothallic (S2C Fig). Within clade D species, we tested *Cryptococcus* sp. 4, where two strains of opposite mating types are available. Although some hyphal growth was observed, it was inconsistent, and no discernible sexual structures formed even after prolonged incubation (S2D Fig). These negative results may reflect low mating efficiency under laboratory conditions, and future genetic approaches such as *CRG1* deletion [141,142] could help uncover latent sexual potential in these lineages. Given the absence of confirmed sexual reproduction in most of the species tested, we resorted to genomic data to examine *MAT* gene content and organization, inferring breeding systems (heterothallic or homothallic) and classifying *MAT* loci configurations as tetrapolar, pseudobipolar, or bipolar, as summarized in Fig 1B.

Consistent with previous studies [34,78], we confirmed that all pathogenic clade A species, including *C. decagattii* and *C. tetragattii*, have a bipolar mating configuration supported by genome assemblies from both mating types. In contrast, most of the nonpathogenic *Cryptococcus* and *Kwoniella* species appeared to be heterothallic and tetrapolar, consistent with the proposed ancestral state of basidiomycetes [9,80,81]. Three exceptions were identified: *Cryptococcus* sp. 3 (clade C), *K. europaea* (clade E), and *K. fici* (clade H) (Fig 1B and 1E). In *Cryptococcus* sp. 3 and *K. fici*, BLAST searches revealed that the key *MAT* genes (*HD* and the pheromone receptor *STE3*) are close together on the same chromosome, consistent with bipolarity (Fig 1E). In contrast, *K. europaea* exhibited a pseudobipolar configuration, with *HD* and *STE3* on the same chromosome but separated by ~8.95 Mb (Fig 1E). These findings prompted detailed characterization of *MAT* loci across species and reconstruction of the chromosomal changes underlying bipolar configurations.

### Structure and evolution of *MAT* loci in *Cryptococcus* and *Kwoniella*

**Identification and general organization of *MAT* loci.** The structure and gene composition of *MAT* loci have been described for several *Cryptococcus* species within clades A, B, and C [16,23,34,35,47,78,82], but in *Kwoniella* analyses have been limited to *K. mangrovensis* and *K. heveanensis*, based solely on fosmid libraries from a single mating type,

leaving the precise structure of *MAT* loci unresolved [66,67]. To expand on these analyses, we combined genome-wide synteny comparisons with gene genealogies to delineate *MAT* loci across both genera. The *HD* locus was defined based on the presence of the *HD1* and *HD2* genes, while the *P/R* locus boundaries were delineated by identifying regions of disrupted synteny between mating types (see Materials and methods).

In tetrapolar *Cryptococcus* and *Kwoniella* species, the *HD* locus is compact, consisting only of the divergently transcribed *HD1*/*SXI1* and *HD2*/*SXI2* genes (S3 Fig). As in other basidiomycetes, mating-type differences are mainly confined to the N-terminal regions of the encoded proteins, and this two-gene structure is strongly conserved. This organization persists in *Kwoniella* species, even though the *HD* locus predominantly resides in subtelomeric regions, inherently more prone to rearrangements (S3J–S3O Fig). By contrast, the *P/R* locus spans a larger chromosomal region and is more variable across species and between mating types of the same species (Figs 2 and S4). It is significantly expanded in tetrapolar *Cryptococcus* species, averaging 91.7 kb compared to 29.3 kb in *Kwoniella* (Figs 2 and S4; S2 Appendix; $P < 0.0001$, Mann–Whitney $U$ test). Within *Cryptococcus*, sizes range from 84.5 kb in *Cryptococcus* sp. 5 (*P/R a2*) to 101.5 kb in *C. floricola* (*P/R a1*), while in *Kwoniella*, they range from 24.3 kb in *K. newhampshirensis* (*P/R a1*) to 34.4 kb in *K. heveanensis* (*P/R a2*) (S2 Appendix).

**Evolution of the *P/R* locus in *Kwoniella*.** Gene content analysis of the *Kwoniella P/R* locus revealed nine genes shared across all species (Figs 1C, 2C, and 2D; S3 Appendix). These include a single pheromone gene (*MFα* or *MFa*) containing the canonical CAAX motif at its C-terminus, a pheromone receptor (*STE3α* or *STE3a*), and two additional genes (*STE20* and *STE12*) with established mating-related roles and found within *MAT* in *Cryptococcus* pathogens [34,83,84]. Other conserved genes include *DBP10* and *REI1*, both implicated in ribosome biogenesis in *Saccharomyces cerevisiae* [85,86]; *ATG7*, an autophagy-related gene [87]; a C2 domain-containing protein (*C2prot*), whose ortholog in *S. pombe* (Ync13) coordinates exocytosis, endocytosis, and cell-wall integrity during cytokinesis [88]; and *NDC80*, encoding a component of the NDC80 kinetochore complex, essential for chromosome segregation and spindle checkpoint activity [89]. In *K. heveanensis* and other clade G species, two additional genes are present within the *P/R* locus: *BPS3*, encoding a protein of unknown function with BTP/PZ and CAP-Gly domains (also present in the *MAT* locus of *Cryptococcus* pathogens), and *UPB6*, a predicted deubiquitinating enzyme associated with the 26S proteasome [90]. Of these, *DBP10*, *REI1*, and *NDC80* are predicted to be essential in *S. cerevisiae* and in *C. neoformans*, based on previous studies [85,86,89,91,92].

Across *Kwoniella*, the *P/R a2* allele is generally larger than the *a1* (Figs 2C, 2D, and S4; S2 Appendix), mainly due to the presence of an additional gene of unknown function. This size difference is also accompanied by truncation of *STE12* within the *a2* allele, with fragments flanking *STE3* on both sides in several species (e.g., *K. mangrovensis* and *K. pini*; Figs 2C and S4). Fragments of the 3′ end of *NDC80* and *BSP3* are likewise found at the right edge of the *P/R a2* allele in certain clade E/F and clade G species, respectively, in addition to apparently intact copies (Figs 2C and S4). These fragments seem to be remnants of structural rearrangements, potentially associated with duplication tracks at inversion breakpoints. Reconstruction of the likely rearrangements leading to the extant *P/R* configuration in *Kwoniella* suggests three distinct inversions have shaped the locus in clades E/F: one involving *STE3* and *STE12*, truncating *STE12* in the *a2* allele; a second relocating *DBP10* and *REI1* from the edge of the locus to its center; and a third moving *NDC80* from the edge to the middle of the locus (S5A Fig). The first two inversions are conserved in clade G, indicating they are ancestral in *Kwoniella*, while clade G additionally underwent rearrangements repositioning *BSP3-NDC80* and *UBP6* to the center of the locus (S5B Fig). This suggests that the *P/R a2* allele has experienced more modifications than the *P/R a1* allele.

Additional complexity is seen in *K. ovata* and *K. endophytica*, where the *P/R* locus contains *STE3a* and *MFa* alleles alongside truncated versions of *STE3α*, with *K. ovata* additionally harboring a truncated *MFα* gene (Fig 3A). This mosaic configuration likely arose through intra-*P/R* recombination, as synteny analyses showed that the left side of the locus resembles that of *P/R a2* strains, whereas the right side aligns more closely with the *P/R a1* (Fig 3A). Gene genealogies further supported this scenario. While deep trans-specific polymorphism has been well-documented

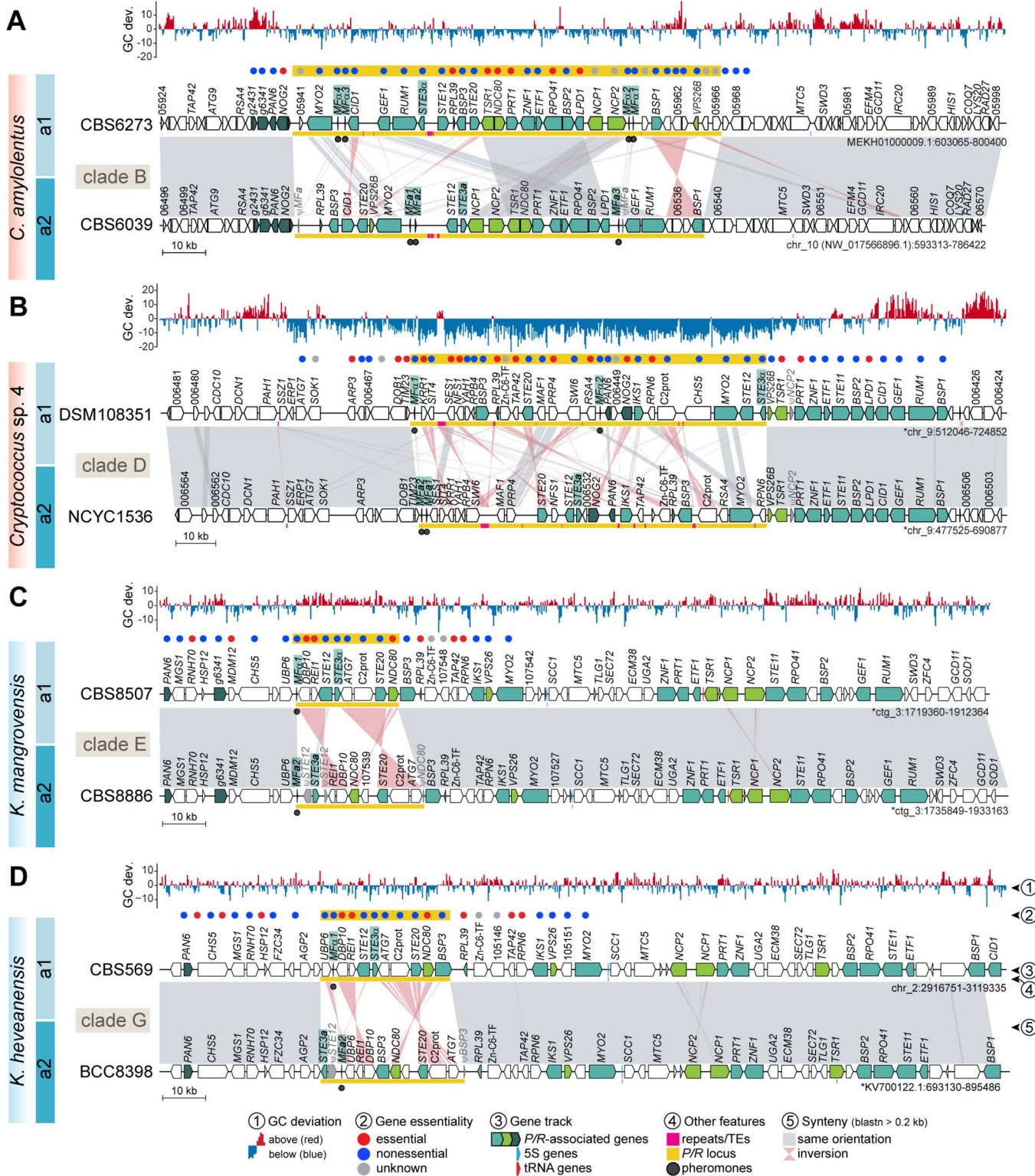

**Fig 2. Synteny analysis of the *P/R* locus in tetrapolar *Cryptococcus* and *Kwoniella* species. (A, B)** Detailed views of the *P/R* locus of representative *Cryptococcus* species from clades B and D. In both clades, the *P/R* locus exhibits lower GC content compared to other genomic regions. In clade D species, the predicted *P/R* locus contains only a subset of the genes found within the *P/R* locus of clade B species or the *MAT* locus of pathogenic

*Cryptococcus* species. The additional genes in clade B or pathogenic species are instead located in the immediate downstream region, suggesting independent expansion of this genomic segment in clade D species. **(C, D)** *P/R* locus region in representative *Kwoniella* species of clades E and G. In *a2* strains, the *STE12* gene appears truncated. Remnants of the *NDC80* gene (in clade E/F species) or the *BSP3* gene (in clade G species) are observed at the right edge of the *P/R a2* allele, likely resulting from inversion events. In all panels, *P/R*-associated genes are colored teal if their orthologs in *Cryptococcus* pathogens are located within the *MAT* locus, with darker teal indicating genes positioned in the flanking regions, and bright green for genes present within the *P/R* locus of *C. amylolentus*, but absent in most *Cryptococcus* pathogens. The *P/R* allele of each strain (*a1* or *a2*) is indicated on the left. Chromosomes inverted relative to their original assembly orientations are marked with asterisks. GC content is depicted as the deviation from the genome average, calculated in 0.5 kb nonoverlapping windows. Gene essentiality within and in the immediate vicinity of the *P/R* locus is inferred from predictions or experimental validations in *C. neoformans* H99 rather than direct assessments in these species. Other features are annotated as shown in the key. See S4 Fig for additional comparisons. The data underlying this figure can be found at https://doi.org/10.5281/zenodo.16987438.

for *STE3*, *STE12*, and *MF* genes, reflecting their ancestral integration into the *P/R* locus across *Cryptococcus* and *Kwoniella* [23,47,67,78], shallower patterns were detected for *NDC80*, *BSP3*, *ATG7*, and *UBP6*. Specifically, *NDC80* exhibited trans-specific polymorphism within clades E/F and G (Fig 3B), while *ATG7*, *BSP3*, and *UBP6* displayed this pattern only within clade G (S5C Fig). Notably, the *NDC80* allele at the right edge of the *P/R* locus in *K. ovata* and *K.*

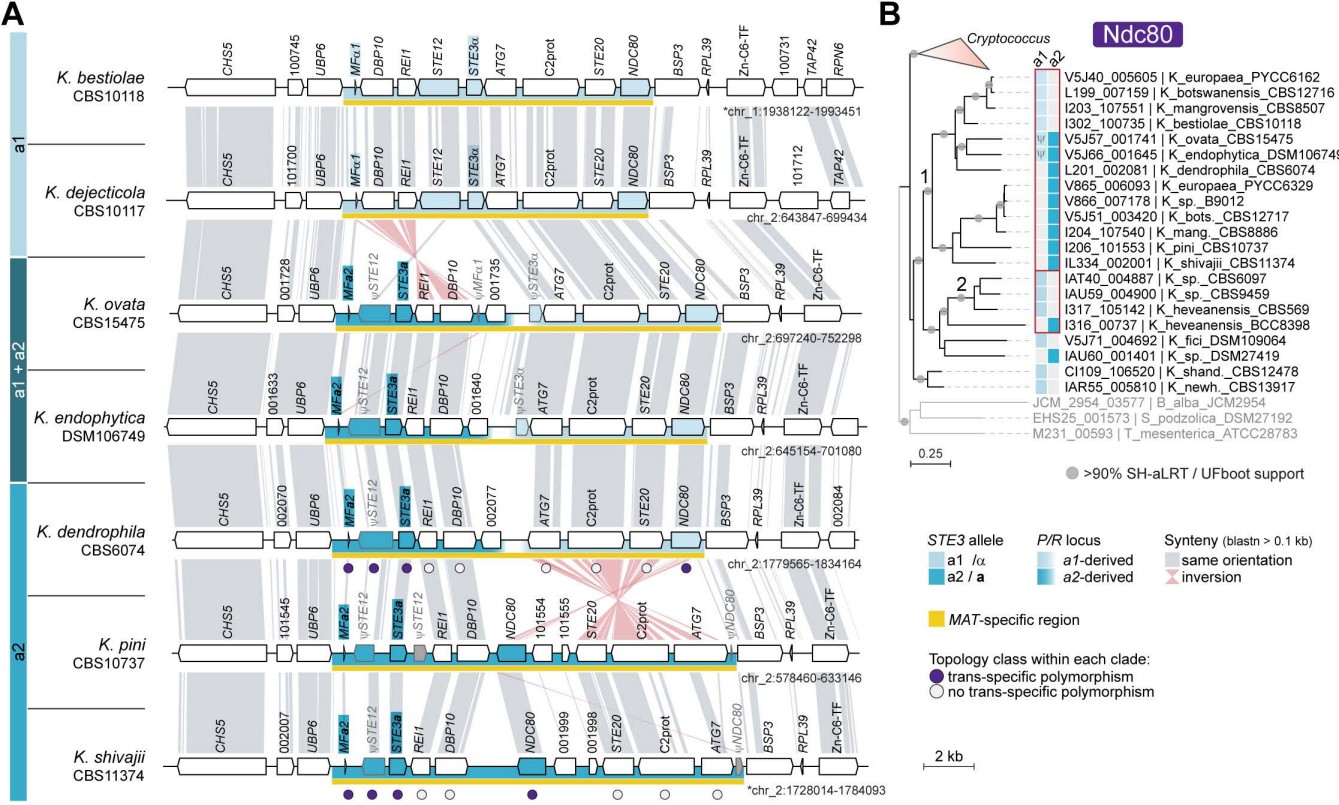

**Fig 3. The *P/R* locus in *K. ovata*, *K. endophytica*, and *K. dendrophila* likely arose through intra-*P/R* recombination. (A)** Synteny analysis comparing *P/R a1* and *P/R a2* alleles in select *Kwoniella* species from clade F. The *P/R* loci of *K. ovata* and *K. endophytica* contain *STE3a* and *MFa* alleles, along with truncated (ψ) versions of *STE3α*. Additionally, *K. ovata* harbors a truncated *MFα* gene. Synteny analysis indicates that the left side of the *P/R* locus in *K. ovata*, *K. endophytica*, and *K. dendrophila* is structurally more similar to *P/R a2* alleles of *K. pini* and *K. shivajii*, while the right side aligns more closely with the *P/R a1* alleles of *K. bestiolae* and *K. dejecticola*. **(B)** Gene genealogy of the *NDC80* gene inferred with IQ-TREE2 (model JTT + G4) showing trans-specific polymorphism across clade E/F species (labeled as 1) and within clade G species (labeled as 2). The *NDC80* alleles of *K. ovata*, *K. endophytica*, and *K. dendrophila* cluster more closely with *a1* alleles than with other *a2* alleles. The data underlying this figure can be found at https://doi.org/10.5281/zenodo.16987438.

PLOS Biology

*endophytica* clustered more closely with *a1* alleles from other species (Fig 3B), and a similar pattern was found in *K. dendrophila*, together pointing to a shared evolutionary history shaped by past intra-*P/R* recombination. This recombination event may have initially produced strains with intercompatible pheromone and receptor pairs, potentially enabling self-filamentous growth. However, subsequent loss of *STE3α* and *MFα* alleles appears to have eliminated self-filamentation. Consistently, solo cultures of *K. ovata* and *K. endophytica* on V8 pH 5 media did not form hyphae, indicating that extant species retain only relics of this ancestral recombination event. Collectively, these findings unveil a dynamic evolutionary history of the *Kwoniella P/R* locus, shaped by structural rearrangements, recombination, and small-scale lineage-specific expansions.

**Evolution and expansion of the *P/R* locus in *Cryptococcus*.** Compared to *Kwoniella*, the *P/R* locus in tetrapolar *Cryptococcus* species is larger, averaging 95.4 kb in clade B and 86.9 kb in clade D, a difference not statistically significant ($P = 0.063$, Mann–Whitney *U* test; Figs 2 and S4; S2 Appendix). It also exhibits more complex rearrangements between mating types, which prevent accurate reconstruction of the evolutionary steps leading to the extant configurations (Figs 2, S4B, and S4E). Gene content analysis revealed striking variation between clades B and D (Fig 1C). Both share a core set of 7 genes (*MF* with two to four copies, *STE3*, *STE12*, *BSP3*, *MYO2*, *STE20*, and *RPL39*; S3 Appendix), but several genes found within the *P/R* locus in clade B species are instead located in the right flanking region of the locus in clade D species (Figs 2A, 2B, S4B, and S4E), and in *Kwoniella* these genes are similarly positioned just outside the locus (Figs 2C, 2D, S4H, S4K, and S4N). Despite only 7 genes being shared between clades B and D, the *P/R* locus is similar in size because clade D species harbor up to 15 distinct genes that are absent from the locus in other *Cryptococcus* or *Kwoniella* clades (Figs 1C, 2B, and S4E). Extending these comparisons across all tetrapolar species of both genera further revealed that only four genes (*MF*, *STE3*, *STE20*, and *STE12*) are conserved within the *P/R* locus, likely reflecting the ancestral state (Fig 1C and 1D; S3 Appendix).

Another notable distinction of the *P/R* locus in *Cryptococcus* is the presence of multiple pheromone genes per mating type [34,47]. Unlike *Kwoniella*, which retains a single pheromone gene per mating type, all heterothallic *Cryptococcus* species carry multiple copies, up to four per mating type. These genes are often arranged in divergent orientations (Figs 2A, 2B, S4B, and S4E), a configuration that may act as inverted repeats. Such an arrangement can facilitate inversion-loop formation during recombination, potentially contributing to structural instability and the expansion of the *P/R* locus boundaries in *Cryptococcus*. Together, these findings support a small ancestral *P/R* locus that expanded independently in descendant lineages by entrapping distinct gene sets through lineage-specific rearrangements.

A final striking observation was the consistently lower GC content of the *P/R* locus compared to genome-wide averages (Figs 2, S4A, and S4D; S4 Appendix). In *C. amylolentus* CBS6039 (clade B), the *P/R* locus has 51.24% GC versus 53.36% genome-wide (*t* test: $t = 96.28$, $P < 0.0001$; permutation test: $P = 0.0303$), with Z-scores ranging from −3.04 to −5.00 across clade B strains. This trend is even more pronounced in clade D species, where the difference can approach 10%. For example, in *Cryptococcus* sp. DSM108351, the *P/R* locus has 49.17% GC compared to a genome-wide average of 59.24% (*t* test: $t = 216.92$, $P < 0.0001$; permutation test: $P = 0.0007$), with Z-scores ranging from −7.56 to −6.71 across clade D strains. To test whether the GC depletion primarily reflects a neutral consequence of reduced GC-biased gene conversion (gBGC) [93,94] in a recombination-suppressed region or also involves relaxed selection on codon usage, we performed four complementary analyses (see S1 Text for details). All pointed to reduced gBGC as the main driver: (i) noncoding GC was significantly lower inside *P/R* than in matched same-chromosome windows, especially in clade D; (ii) coding sequences exhibited a marked shift toward AT-ending codons; (iii) codon-bias indices showed that *P/R* codon usage closely matched expectations based on nucleotide composition, unlike background genes; and (iv) codon adaptation index (CAI) differences were not significant after controlling for GC3 and gene length.

## Independent origins and evolutionary outcomes of *MAT* locus fusion

**Post-fusion structural dynamics of *MAT* in *Cryptococcus* pathogens.** The fusion of the *P/R* and *HD* loci into a single nonrecombining *MAT* locus is well established in clade A, which encompasses all human-pathogenic *Cryptococcus* species. We extended *MAT* locus analysis to all recognized species in this clade (Figs 1, 4, S6, and S7; S1 Appendix). For *C. deneoformans*, our dataset included congenic strains JEC21α and JEC20**a**, their progenitor strains NIH12α and NIH433**a**, and an additional *MAT***a** strain (NIH430) (S6 Fig and S1 Appendix). For *C. neoformans*, we examined multiple strains of both mating types across the four main lineages (VNI, VNII, VNBI, and VNBII), including eight newly sequenced genomes assembled telomere-to-telomere (T2T) (Fig 4A and S1 Appendix).

Synteny comparisons revealed that the *MAT***a** allele is generally longer than the *MAT*α allele in both *C. neoformans* ($P < 0.0001$, Mann–Whitney *U* test) and the *C. gattii* species complex ($P = 0.04142$, Mann–Whitney *U* test; Fig 4C and S2 Appendix), with a similar trend in *C. deneoformans*, although limited sampling precluded statistical support. The *MAT***a** allele also exhibits greater structural diversity than *MAT*α, as seen in the *C. gattii* complex (S7A Fig) and in *C. neoformans*, where frequent inversions were detected even among *MAT***a** strains of the same VN lineage (e.g., VNI strains Bt130 versus IUM96-2828, or strain Bt206 compared to other VNBII strains; Fig 4D). Inversion breakpoints were often associated with pheromone genes, consistent with the hypothesis that multiple pheromone copies may act as mediators of structural changes. In contrast, *MAT*α is structurally more conserved with only limited inversions involving the pheromone–gene pairs flanking the *PRT1* and *ZNF1* genes (Figs 4D and S7). Comparison of the entire *MAT* chromosome across *C. neoformans* strains further showed that most of the intraspecific structural variation is confined to the *MAT* locus, with variation outside *MAT* largely restricted to the inherently dynamic centromeric and telomeric regions (Fig 4B).

As part of this analysis, we also detected significant structural divergence between the *MAT***a** alleles of *C. neoformans* VNBI (Ftc555-1 and Bt63) and VNI strains (Fig 4D). The VNI lineage has been hypothesized to have diverged initially with only the *MAT*α allele, possibly due to a bottleneck associated with a small founding population, and later acquired the *MAT***a** allele via introgression from a VNBI *MAT***a** donor [95]. To date, only five VNI *MAT***a** strains have been identified [95–100], three of which are now represented by high-quality genome assemblies (125.91 and Bt130, sequenced in this study, and IUM96-2828, sequenced previously [101] and annotated here; Fig 4D). This structural divergence did not contradict the introgression model but raised questions about how such differences could be reconciled with an introgressed origin.

To revisit this question with improved data, we reanalyzed SNP data leveraging newly generated reference genomes. Genome-wide SNP analysis, with VNBI *MAT***a** strain Ftc555-1 as the reference, revealed markedly lower SNP densities within the *MAT***a** locus in VNI strains compared to VNBII strains (Fig 5A and 5B). Conversely, across the rest of the genome, VNI *MAT***a** strains diverged more strongly from the VNBI reference, while VNBII strains showed lower SNP densities (Fig 5A and 5B). Notably, the typically high SNP density between VNI and VNBI strains was restored immediately downstream of *MAT* and ~5 genes upstream, consistent with recombination hotspots flanking the locus [102] and previously mapped introgression boundaries [95]. Phylogenetic analyses reinforced these patterns: VNBI strains clustered more closely with VNBII strains in trees inferred from genome-wide and chr. 5-specific SNPs (Fig 5C and 5D), whereas *MAT*-specific SNP trees placed VNI and VNBI *MAT***a** strains together, consistent with a more recent shared ancestry for the *MAT***a** allele (Fig 5E). Together, these findings refine the VNBI-to-VNI introgression model (Fig 5F) and indicate that lineage-specific rearrangements following introgression account for the structural differences observed.

In addition to introgression, other forces also shaped *MAT* architecture in *C. neoformans*. Species-specific gene losses (e.g., *NCP1* and *NCP2*, which are absent in most species) and TE insertions contributed further to allele differentiation (S7 Fig). TE activity appears especially important, as the *C. deneoformans MAT* locus contains over five times more TEs than genome averages outside centromeres [16,34,103]. In line with this, we identified novel insertions of two distinct KDZ transposons, recently characterized in *C. neoformans* as large mobile elements (~11 kb) with terminal inverted repeats

**Fig 4. Synteny analysis of the mating-type locus (*MAT*α and *MAT***a**) across different *Cryptococcus neoformans* lineages. (A)** Phylogenetic tree of representative *C. neoformans* strains from different VN lineages, inferred using maximum likelihood analysis of a concatenated matrix of protein alignments from 5,439 single-copy genes. Branch support is based on SH-aLTR and UFBoot tests. Mating types were determined based on the allelic

version of the pheromone receptor gene (*STE3*). **(B)** Synteny analysis of chr. 5, where the *MAT* locus resides, showing significant conservation across strains, except within the *MAT* and centromere regions, which exhibit rearrangements. Strains are organized by mating type and lineage, as in panel **D**. **(C)** Box plot comparing *MAT* locus size between α and **a** strains of *C. neoformans* (*C.n.*) and *C. gattii* species complex (*C.g.c.*). The red dashed line, blue line, and boxes denote the mean value, median value, and interquartile range, respectively. Outliers are labeled with the corresponding strain name. Statistical significance (*P*-values) was determined by the Mann–Whitney *U* test. **(D)** Detailed synteny of the *MAT* locus, highlighting strain- and mating-type-specific rearrangements. The *MAT***a** configuration is more variable both across and within VN lineages, existing in three distinct configurations, whereas the *MAT*α structure is more conserved, with limited rearrangements between ancestrally linked homeodomain (*HD*) and pheromone receptor (*P/R*) genes. Observed rearrangements appear to result from inversions mediated by identical copies of pheromone genes. KDZ transposons are detected within the *MAT* locus of certain strains. Genes typically associated with the *P/R* and *HD* loci in tetrapolar species are colored green and gold, respectively, with darker shades denoting genes flanking the *MAT* locus in *Cryptococcus* pathogens. Gene essentiality within the *MAT* locus, as predicted or experimentally validated in strain H99, is indicated by color-coded circles as given in the key. The data underlying this figure can be found at https://doi.org/10.5281/zenodo.16987438.

and target site duplications [104]. *KDZx* was present in strains 125.91, C45, PMHc1023, and T4, while *KDZ1* was found in strain Ze90-1 (Fig 4D). Altogether, these findings underscore the dynamic evolution of the *MAT* locus in *Cryptococcus* pathogens, shaped by an interplay of structural rearrangements, gene loss, TE activity, and introgression.

**Independent fusion of *P/R* and *HD* loci in *Cryptococcus* sp. 3.** Our initial screening of *MAT* gene content revealed that the *P/R*- and *HD*-associated genes are adjacent on the same chromosome in *Cryptococcus* sp. 3, indicating a fusion of these two regions (Fig 1E). While only a single strain (CMW60451) is available, limiting precise delineation of *MAT* boundaries, the minimum estimated size is ~90 kb based on the distance between the two most distal pheromone genes (Fig 6C). If boundaries are extended to include additional genes that are part of the *MAT* locus in pathogenic *Cryptococcus*, the locus could span ~120 kb, from *RPL22* on the left to *STE12* on the right (Fig 6C). Under this broader definition, 15 genes are shared between *Cryptococcus* sp. 3 and clade A species (Figs 1C and 6C; S3 Appendix), including four *MF*α pheromone gene copies and three genes typically associated with the *HD* locus (*RPL22*, *CAP1*, and *SPO14*). Additionally, two regions within the *MAT* locus contain TEs: one with a complete KDZ-like transposon and another with only remnants of this element (Fig 6C).

Further analysis uncovered an unusual combination of *MAT* alleles. Unlike the bipolar *Cryptococcus* pathogens, where *MAT*α strains carry *HD1* (*SXI1*) linked to *STE3*α and *MAT***a** strains have *HD2* (*SXI2*) linked to *STE3***a**, *Cryptococcus* sp. 3 retains only *SXI2* linked to *STE3*α. Although this distinct configuration suggested an independent fusion event at the origin of the *MAT* locus in *Cryptococcus* sp. 3, the unresolved phylogenetic placement of clade C relative to clade A left open the possibility that the two clades share a more recent common ancestor. In this scenario, the fusion of the *P/R* and *HD* loci could have occurred in their shared ancestor rather than independently in both lineages.

To test this hypothesis, we analyzed the phylogenetic clustering patterns of genes within the predicted *MAT* region shared with pathogenic *Cryptococcus*. Genes ancestrally present in the *P/R* locus, such as *STE3*, *STE12*, and *MYO2*, were excluded from the analysis because they exhibit deep trans-specific polymorphism predating the divergence of *Cryptococcus* species [23,47,67,78], making them uninformative. Instead, we focused on four genes (*BSP3*, *PRT1*, *SPO14*, and *CAP1*), which are part of *MAT* in pathogenic species and display mating-type-specific clustering within them [34]. The analysis revealed that *Cryptococcus* sp. 3 sequences do not group with either the **a** or α allele-specific clusters from pathogenic species; rather, they are distinct and fall outside these clusters, which is more consistent with an independent fusion event than one inherited from a shared ancestor (S8 Fig).

Having established from gene phylogenies that the *MAT* fusion in *Cryptococcus* sp. 3 likely arose independently, we next examined chromosome-scale structural changes underlying the *P/R*-*HD* linkage by reconstructing synteny blocks with SynChro [105]. *K. shandongensis* was selected as reference because (i) it is tetrapolar, with *P/R* and *HD* loci on separate chromosomes representing the ancestral state (Figs 1, S3N, and S4N); (ii) *Kwoniella* species show significantly fewer interchromosomal rearrangements than *Cryptococcus*, likely due to their simpler, smaller centromeres and lower TE density, as both high TE content and large, complex centromeres promote genomic rearrangements in *Cryptococcus*

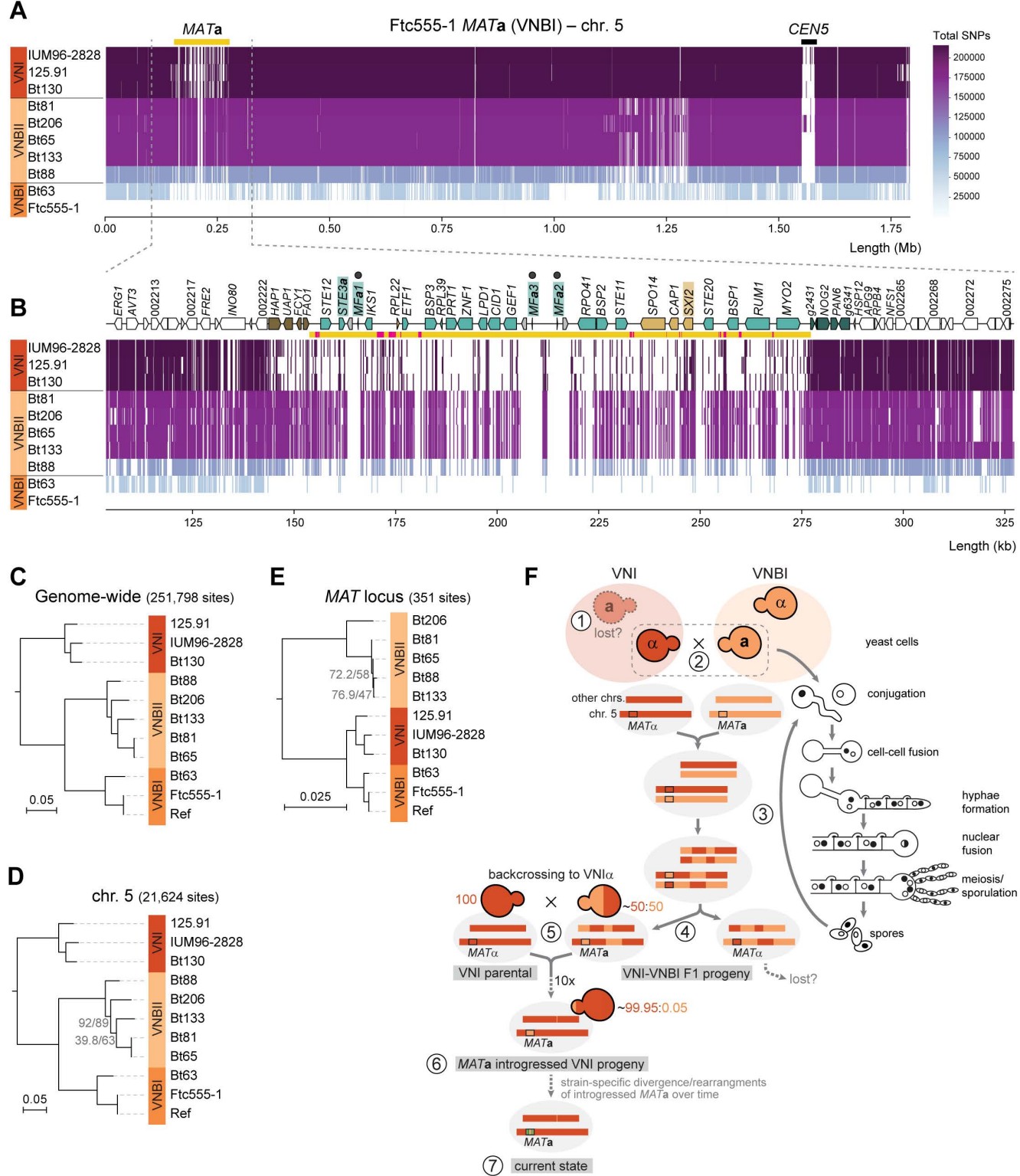

**Fig 5. SNP distribution, phylogenetic analysis, and revised introgression model of the *MAT*a allele from VNBI into VNI. (A)** Genome-wide distribution of SNPs across chromosome 5 in *C. neoformans MAT*a strains from different VN lineages relative to the *C. neoformans* Ftc555-1 VNBI genome reference. Empty spaces in the plots represent either sites identical to the reference genome (monomorphic sites) or regions excluded due to

low coverage or repetitive/duplicated sequences. Strains on the y-axis are ordered by the total number of SNPs across the genome, with a color gradient (right) indicating SNP counts: darker colors represent higher SNP densities relative to the reference (strains positioned at the top), while lighter colors indicate fewer SNPs (strains at the bottom). Reads from the reference strain were also mapped as a control. The x-axis shows genomic coordinates. **(B)** Zoomed-in view of the *MAT* locus region plus 50 kb flanking regions, with genes and other features represented as in [Fig 4]. Notably, VNI strains exhibit a markedly lower number of SNPs at the *MAT***a** locus relative to the VNBI reference, compared to VNBII strains. **(C)** Midpoint-rooted tree based on genome-wide SNPs across all chromosomes, illustrating the evolutionary relationships of the strains. **(D)** Midpoint-rooted tree constructed using SNPs from chromosome 5 only, showing similar relationships to the genome-wide analysis. **(E)** Midpoint-rooted tree based on SNPs restricted to the *MAT* locus, highlighting a closer relationship between VNI and VNBI strains. In all trees, branches are shown as the number of substitutions per site (scale bars) and have >95% support based on SH-aLTR and UFBoot tests, unless otherwise indicated. "Ref" represents the reference genome assembly employed for read mapping and is thus identical to Ftc555-1 reads, serving as a control to validate the SNP calling pipeline. **(F)** Proposed model for the introgression of the VNBI *MAT***a** allele into VNI, shown alongside the key stages of the *C. neoformans* sexual cycle (right). Step 1: possible loss of the *MAT***a** allele in the VNI lineage, likely due to a bottleneck; step 2: mating and cell-cell fusion between a VNBI (*MAT***a**) and a VNI (*MAT*α) strain; step 3: formation of dikaryotic hyphae, nuclear fusion, meiosis, recombination, and sporulation; step 4: some of the resulting F1 progeny emerged with a ~50:50 VNBI:VNI genome composition, retaining the VNBI *MAT***a** locus; step 5: repeated backcrossing of the *MAT***a** F1 progeny to VNI *MAT*α strains progressively purging VNBI genomic regions while retaining the introgressed *MAT***a** allele; step 6: after ~10 backcrosses, the genome became ~99.95% VNI, with only the VNBI *MAT***a** locus and possibly a small fraction (~0.05%) of the VNBI genome remaining; step 7: over time, the introgressed VNBI *MAT***a** locus diverged and underwent rearrangements within the VNI background, reflecting strain-specific changes. The data underlying this figure can be found at https://doi.org/10.5281/zenodo.16987438.

[49,106–108]; and (iii) it retains a 14-chromosome karyotype, which is the predicted ancestral state for both genera [49] (Figs 6 and S9A).

The nuclear genome of *Cryptococcus* sp. 3 comprises 13 chromosomes (S1 Appendix), representing a reduction from the ancestral 14-chromosome karyotype. While we hypothesized that this reduction could explain the *P/R-HD* fusion, examination of the lost centromere showed it did not involve the *MAT*-containing chromosome (detailed in S2 Text and S9B Fig). Thus, physical linkage of the *P/R* and *HD* loci in *Cryptococcus* sp. 3 arose independently of the karyotypic reduction. We therefore compared the *MAT*-containing chromosome of *Cryptococcus* sp. 3 to the *P/R*- and *HD*-bearing chromosomes of *K. shandongensis*. Synteny analysis revealed significant alignment between chr. 11 of *Cryptococcus* sp. 3 and large regions of *K. shandongensis* chrs. 12 and 5, which carry *P/R* and *HD*, respectively (Fig 7A). Additionally, a small segment from the *P/R* chromosome is embedded towards the left end of chr. 11 within a region otherwise derived from the *HD* chromosome. These patterns suggest that the physical linkage likely began with a translocation that brought the two loci onto the same chromosome, initially far apart, followed by an inversion relocating the *HD* locus closer to the *P/R* locus, resulting in their fusion (events 1–3 in Fig 7A). Notably, the centromere-flanking regions of chr. 11 in *Cryptococcus* sp. 3 show considerable synteny to the *CEN12*-flanking region in the *K. shandongensis* *P/R* chromosome, indicating a shared evolutionary origin and suggesting that the translocation did not involve centromeric recombination but occurred elsewhere (Fig 7A).

**Revisiting the model of *P/R–HD* loci linkage in *Cryptococcus* pathogens.** Prompted by these findings, we revisited the proposed model for the physical linkage and subsequent fusion of *P/R* and *HD* loci in *Cryptococcus* pathogens. This model, originally devised from comparisons between *C. amylolentus* (tetrapolar; clade B) and *C. neoformans* (bipolar; clade A), posited that the physical linkage was initiated by a chromosomal translocation mediated by ectopic recombination between repetitive elements within centromeric regions of *P/R*- and *HD*-containing chromosomes. This translocation would place the two *MAT* loci on the same chromosome, initially separated by the centromere. Subsequent rearrangements, such as inversions or transpositions, were hypothesized to bring the two loci closer together, ultimately resulting in their fusion and establishing the contiguous *MAT* locus of *Cryptococcus* pathogens [35,37].

To reassess this hypothesis, we compared multiple *Cryptococcus* species spanning the four clades, focusing on synteny and chromosomal architecture. In *C. neoformans*, the *MAT* locus resides on chr. 5. We found that its centromere (*CEN5*) shares conserved synteny on both flanking regions with *CEN12* of *K. shandongensis*, as well as with *CEN11* of *Cryptococcus* sp. 3 (clade C) and *CEN9* of *Cryptococcus* sp. DSM108351 (clade D) (Fig 6B). These conserved synteny patterns indicate a shared evolutionary origin for these centromeres and strongly suggest that they were not directly

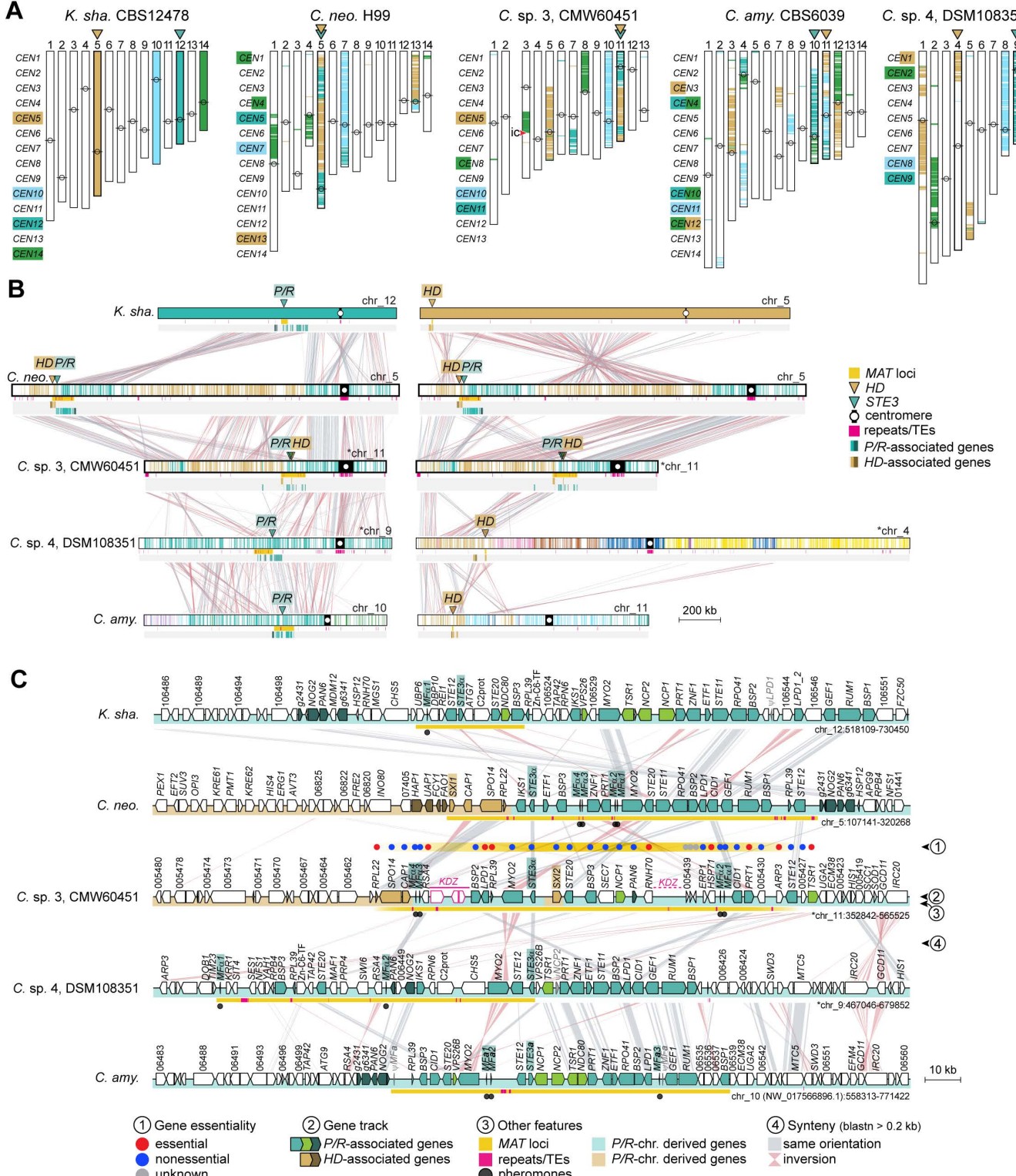

**Fig 6. Transition to linked *P/R-HD* loci in *Cryptococcus* sp. 3 (CMW60451) and Cryptococcus pathogens. (A)** The karyotype of *K. shandongensis* (with 14 chrs.) served as the reference for reconstructing synteny blocks in pairwise comparisons. For simplicity, only synteny blocks corresponding to *P/R*- and *HD*-containing chromosomes (chrs. 12 and 5, colored teal and gold, respectively) were plotted in representative *Cryptococcus* species of clades

A (*C. neo.* H99), B (*C. amy.* CBS6039), C (*C.* sp. CMW60451), and D (*C. sp.* DSM108351). Besides the *MAT*-containing chromosomes, synteny blocks of two additional chromosomes in *K. shandongensis* (chrs. 10 and 14), which correspond to a large portion of *P/R* and *HD* chromosomes in *C. amylolentus* were also plotted (color-coded light blue and green, respectively). Other synteny blocks can be visualized in S9A Fig. A red arrowhead pinpoints the predicted location of an inactivated centromere (ic) in *Cryptococcus* sp. CMW60451. **(B)** Linear chromosome plots depicting gene synteny conservation across species with zoomed-in views depicted in **(C)**. In panel B, the *MAT* chromosome of bipolar species (*C. neoformans* and *C.* sp. 3, CMW60451) is shown twice: the left column depicts synteny relative to the *P/R*-containing chromosomes of tetrapolar species, and the right column depicts synteny relative to the *HD*-containing chromosomes.

involved in the translocation linking the *P/R* and *HD* loci in *C. neoformans*. In contrast, the *P/R* chromosome (chr. 10) of *C. amylolentus* (clade B) shows centromere-flanking regions that align with two distinct chromosomes, consistent with intercentromeric recombination and pointing to a derived rather than ancestral centromere structure.

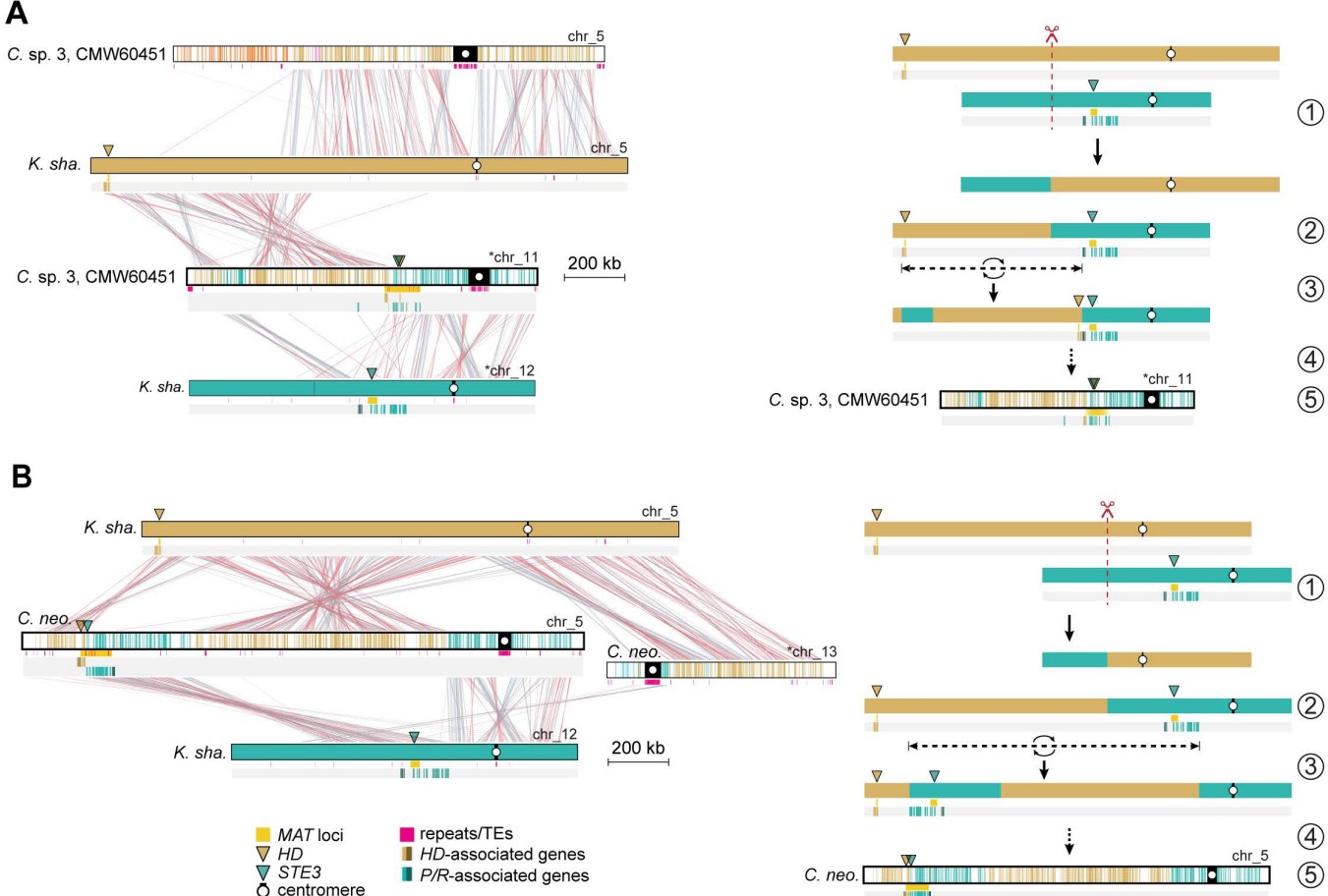

**Fig 7. Hypothesized mechanisms underlying the tetrapolar-to-bipolar transitions in *Cryptococcus* sp. 3 and in Cryptococcus pathogens.** Comparative synteny analysis between *K. shandongensis* and **(A)** *Cryptococcus* sp. 3 or **(B)** *C. neoformans*, illustrating the inferred sequence of chromosomal rearrangements leading to this transition. In both panels, the left side displays pairwise alignments, with syntenic regions connected by red (collinear) and gray (inverted) links, while the right side depicts a stepwise model of the proposed structural changes. The transition likely began with (1) an initial chromosomal break and reciprocal translocation (red dashed line), leading to (2) the repositioning of the *P/R* locus onto the same chromosome as the *HD* locus. (3) In *Cryptococcus* sp. 3, a subsequent inversion relocated the *HD* locus closer to the presumed ancestral position of the *P/R* locus, whereas in *Cryptococcus* pathogens, an inversion instead shifted the *P/R* locus toward the original location of the *HD* locus. (4) Additional structural modifications further refined the newly linked configuration, ultimately resulting in (5) the extant *MAT* locus organization observed in each species. These models suggest that despite independent evolutionary trajectories, both lineages underwent convergent genomic rearrangements that facilitated the transition to bipolar mating systems.

Comparisons between *C. neoformans* and *K. shandongensis* further suggest a model in which the physical linkage and subsequent fusion of *P/R* and *HD* in *C. neoformans* arose through a mechanism similar to that proposed for *Cryptococcus* sp. 3, involving a chromosomal translocation outside the centromere followed by an inversion bringing the two loci into closer proximity (Fig 7B). In *C. neoformans*, however, the inversion seems to have relocated the *P/R* locus towards the *HD* locus near the left chromosomal end (event 3 in Fig 7B). These broader analyses refine the earlier model and suggest that the physical linkage of *P/R* and *HD* loci in *Cryptococcus* species likely arose through alternative mechanisms, independent of centromeric recombination.

**Tetrapolar to bipolar and pseudobipolar transitions in *Kwoniella*.** We recently reported that karyotype reduction within *Kwoniella* occurred progressively and independently, often with formation of "giant" chromosomes (up to 18 Mb) through repeated fusions [49]. Although such events could potentially lead to *P/R-HD* physical linkage, only two *Kwoniella* species harbor both loci on the same chromosome (Fig 1B). In *K. europaea* (clade E), the *HD* and *P/R* loci are located on the same chromosome but remain far apart (~8.95 Mb) (Figs 1E and S4G), representing a pseudobipolar configuration. This arrangement likely arose from a lineage-specific rearrangement, in which the progenitor "giant" chr. 1 of clade E species underwent a translocation with chr. 2 [49]. Given their physical distance, the two *MAT* loci are still expected to recombine.

In *K. fici*, by contrast, the *HD* locus lies less than 13 kb from the *P/R* gene cluster, both located near the end of chr. 1 (Fig 8D). This "giant" chromosome likely arose independently through fusion events, similar to those documented in *Kwoniella* clades E and G [49]. After divergence from clade I (*K. shandongensis*/*K. newhampshirensis*), the chromosomes corresponding to chrs. 11 and 12 of *K. shandongensis* fused in the descendant lineage, producing the ancestral *P/R* chromosome, which persists as a single chromosome in most *Kwoniella* species (e.g., chr. 2 in *Kwoniella* sp. 4 DSM27419; Fig 8A) [49]. In *K. fici*, however, this ancestral fusion product fragmented further: one block now resides at the terminal region of *K. fici* chr. 1, while another segment lies on chr. 4 (Fig 8C). Similarly, most of the *K. shandongensis*/*Kwoniella* sp. 4 *HD* chromosome (chr. 5), aligns with the opposite end of *K. fici* chr. 1, while a small portion is on chr. 4, adjacent to the *P/R*-derived block. Only a few key *HD*-associated genes (*SXI1*, *SXI2*, and *RPL22*) are positioned near the *P/R* locus gene cluster at the right end of chr. 1 (Fig 8D). Notably, this derived bipolar arrangement retains both *HD* genes, unlike bipolar *Cryptococcus* species where opposite mating types carry only one. Overall, these observations indicate that complex translocations united blocks from ancestral *P/R* and *HD* chromosomes, producing the unique fused configuration in *K. fici*.

## Analysis of sexual reproduction in *Kwoniella mangrovensis*

*Kwoniella mangrovensis* (clade G) can reproduce sexually in the lab [52,66] and has only three chromosomes. To better understand its reproductive biology, we analyzed recombination, ploidy variation, and the potential for self-fertility in the progeny. The two originally sequenced strains, CBS8507 (*P/R a1*, *HD b1*) and CBS8886 (*P/R a2*, *HD b1*), share identical *HD b1* alleles, making them incompatible for mating (Fig 9A). However, Guerreiro and colleagues [66] reported a third strain, CBS10435, as haploid and compatible with CBS8507. We sequenced CBS10435 with Illumina and confirmed it carries the *P/R a2* allele and a distinct *HD b2* allele, rendering it compatible with CBS8507 (Fig 9A).

Crosses between CBS8507 and CBS10435 on CMA and V8 pH 5 media produced basidia with characteristic globose and lageniform morphologies, consistent with prior descriptions (Fig 9B) [52]. Progeny were recovered by dissecting eight clusters of cells germinating from basidia along the hyphae (S5 Appendix). In total, 51 F1 progeny were obtained from both culture media. To avoid analyzing clonally derived genotypes, we screened for recombination with a PCR-RFLP approach using two markers per chromosome, positioned on either side of the centromere (S5 Appendix). Evidence of allele exchange and heterozygosity was detected, and nine F1 progeny with distinct genotypes were selected for whole-genome sequencing. SNPs between the parental strains served as markers to infer inheritance patterns associated with recombination and loss of heterozygosity (LOH), while read coverage analysis and FACS were combined to assess ploidy.

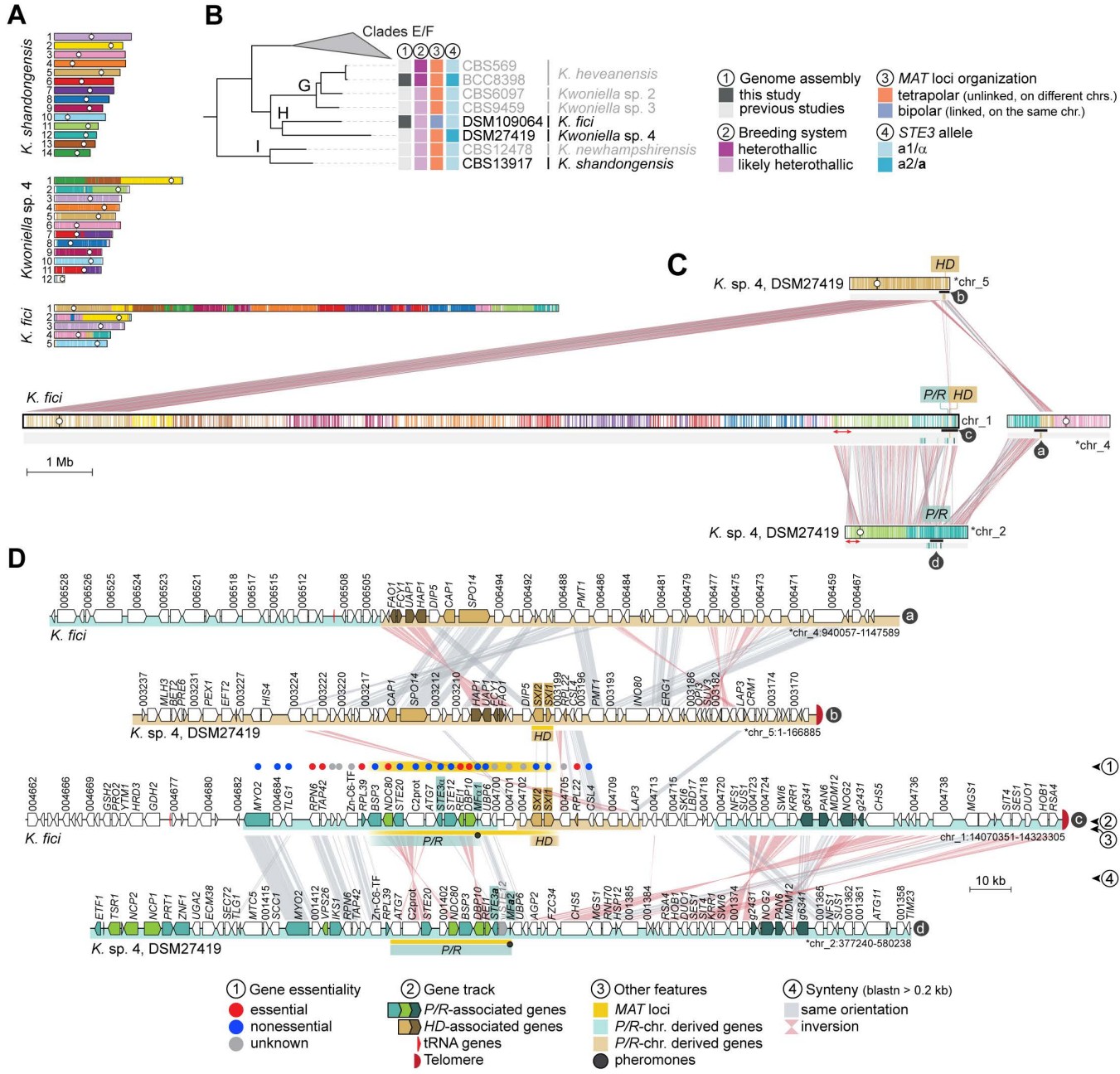

**Fig 8. Linkage of *P/R* and *HD* loci in *Kwoniella fici* and *MAT* locus structure. (A)** The karyotype of *K. shandongensis* (with 14 chrs.) served as a reference for reconstructing synteny blocks in pairwise comparisons with *Kwoniella* sp. 4 and *K. fici*. **(B)** Phylogenetic relationships among selected *Kwoniella* species, highlighting their *MAT* locus organization and inferred breeding systems. **(C)** Synteny comparison between *K. fici* and its closest relative *Kwoniella* sp. 4, illustrating chromosomal rearrangements underlying the linkage of *P/R* and *HD* loci in *K. fici*. A red double-headed arrow indicates that the centromere-proximal regions of chr. 2 in *Kwoniella* sp. 4 correspond to a region near the fusion point on the *K. fici* "giant" chromosome (chr. 1), whereas the telomere-proximal regions of *Kwoniella* sp. 4 chr. 2 align with more internalized regions. This suggests that a large pericentric inversion targeting the centromere-adjacent region is associated with this fusion event, as previously reported for other *Kwoniella* species [49]. **(D)** Detailed gene-level organization of the *MAT* locus in *K. fici* compared to *Kwoniella* sp. 4. Each track (labeled a–d in dark circles) corresponds to a specific region marked by a black bar in panel **C**. Chromosomes inverted relative to their original assembly orientations are marked with asterisks.

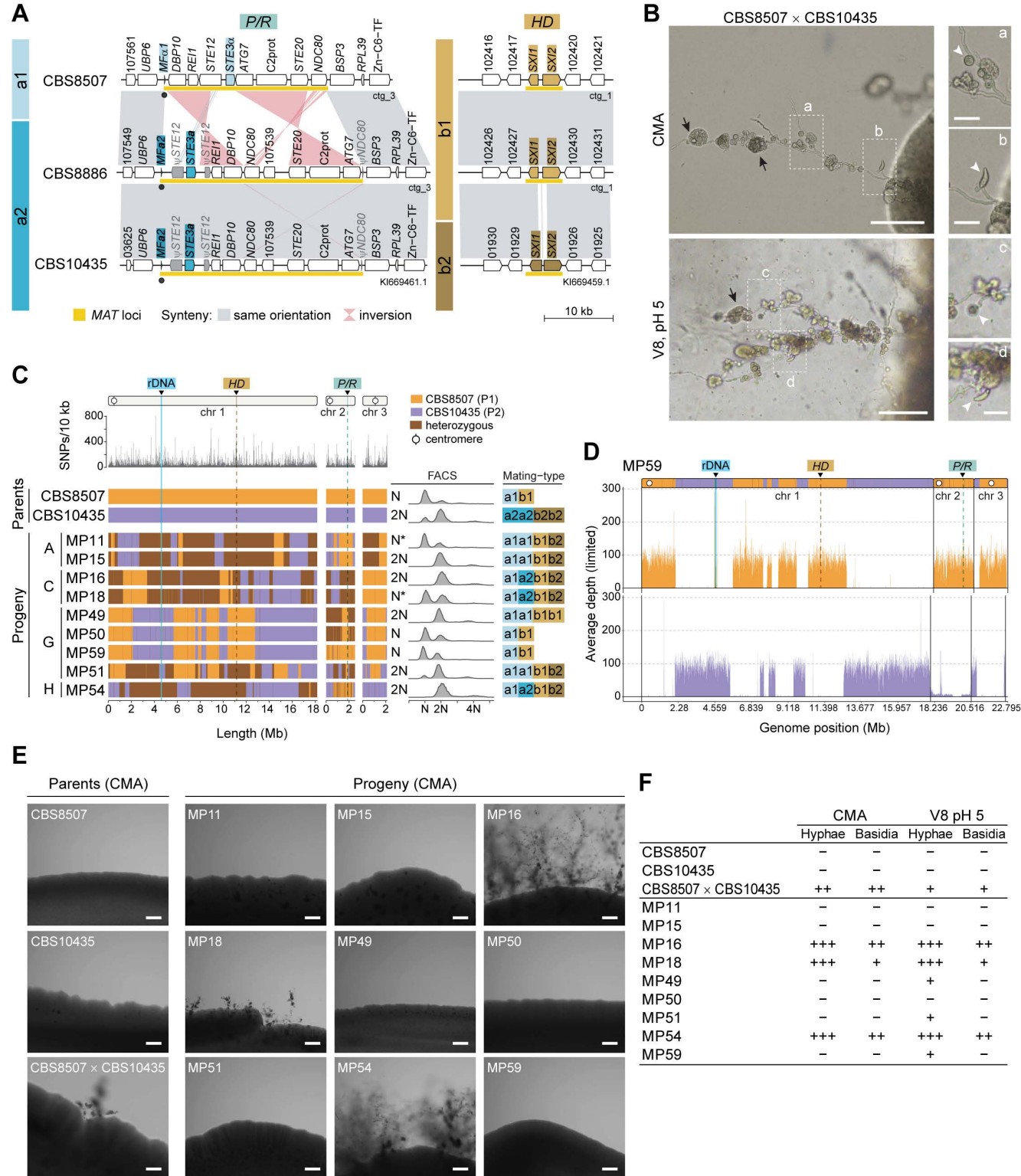

**Fig 9. Sexual reproduction and recombination in *Kwoniella mangrovensis*.** (**A**) Synteny analysis of the *P/R* and *HD* loci among three *K. mangrovensis* strains of different mating types (CBS8507, *a1b1*; CBS8886, *a2b1*; and CBS10435, *a2b2*). Note the pronounced divergence in the N-terminal domains of both *HD1* (*SXI1*) and *HD2* (*SXI2*) gene products, which are known to mediate mating-type specificity in other basidiomycetes. (**B**)

Micrographs showing hyphal filaments extending from colony peripheries in a sexual cross between CBS8507 and CBS10435 after 2 weeks of incubation on CMA and V8 pH 5 media, in the dark, at room temperature. Insets depict two types of basidia: globose (a–c) and lageniform (b–d). Black arrows indicate clusters of cells emerging near basidia (either at the surface or embedded), scored as potential meiotic progeny (basidiospores). Single cells from these clusters were isolated using a micromanipulator, cultured into colonies, and genotyped by PCR-RFLP (see S5 Appendix). Putative recombinants were further analyzed using Illumina sequencing. Scale bars = 100 μm (25 μm in insets). (C) Genotypic analysis of selected meiotic progeny. SNP density between the parental strains was calculated as the number of SNPs per 10 kb (top). The genotypes of 9 segregants, derived from four distinct cell clusters, were inferred from SNP data and are depicted as follows: orange for regions inherited from CBS8507, purple for regions inherited from CBS10435, and brown for heterozygous regions (i.e., inherited from both parents). Instances of recombination or loss of heterozygosity (LOH) are detected by changes in genotype along the chromosomes. Discrepancies in ploidy, as inferred from FACS and sequencing read coverage, suggest potential genomic instability (marked by asterisks). Mating-type identity was inferred through sequencing, coverage analysis, and FACS. (D) Sequencing coverage plot for progeny MP59 with color-coded contributions from each parent. Haplotype blocks inferred from SNP data are overlaid for each chromosome for comparison (additional progeny data is presented in S8 Fig). (E) Self-filamentation phenotype of *K. mangrovensis* progeny. Parental strains CBS8507 and CBS10435 (grown individually and in co-culture) and their recovered progeny were cultivated on CMA at room temperature. Self-filamentation was assessed after 2 weeks of incubation. While neither parental strain exhibited self-filamentation in solo culture, their co-culture produced hyphal filaments and basidia. Progeny MP16, MP18, and MP54 also exhibited robust self-filamentation on both CMA and V8 pH 5 (see S9 Fig), whereas progeny MP49, MP51, and MP59 displayed weaker filamentous growth, detectable only on V8 pH 5 (S9 Fig). Scale bars = 200 μm. (F) Summary of the growth phenotype of the recovered progeny. The production of mycelium and basidia was classified as: extensive (+++), when observed across the entire periphery of the mating patch; moderate (++), when restricted to specific areas of the mating patch; poor (+), when limited to a single spot of the mating patch and slower to develop; and negative (−), when no filamentous growth was observed. Results represent observations from two independent tests.

These analyses revealed that most F1 progeny were nonhaploid (Figs 9C, 9D, S10, and S11). Genotyping and FACS results were largely concordant, though some isolates (e.g., MP11 and MP18) showed discrepancies, suggesting variability within cell populations, possibly stemming from intrinsic genomic instability. Accordingly, mapping reads to a combined reference genome (CBS8507 and CBS10435) revealed uneven coverage for certain chromosomes. For example, CBS10435 chr. 2 exhibited low coverage in progeny MP59 (Fig 9D). Although CBS8507 and CBS10435 were previously reported as haploid [66], our FACS analysis showed that CBS10435 is predominantly diploid with a haploid subpopulation (Figs 9C and S11). This difference in parental ploidy likely underlies the aneuploidy observed in the progeny, as haploid–diploid matings are prone to chromosome mis-segregation, and the reduced karyotype of *K. mangrovensis* (3 chromosomes) may further amplify these effects.

Analysis of the *MAT* gene content revealed that some progeny carried fully compatible *P/R* and *HD* alleles (*a1a2b1b2*), potentially enabling self-fertility (Fig 9C). To investigate this, we cultured the F1 progeny under mating conditions and compared their phenotypes to a parental cross (Figs 9E, 9F, and S12). On CMA and V8 pH 5 media, progeny MP16, MP18, and MP54 with fully compatible *MAT* alleles produced hyphae and basidia similar to the parental cross. Some hyphal growth was also observed in solo culture of strains MP49, MP51, and MP59, exclusively on V8 pH 5, but the filaments were shorter and had no structures resembling basidia (S12 Fig). This morphology resembled that of previously described self-filamentous *K. mangrovensis* strains, which form irregular, highly branched hyphae without clamp cells [66]. Taken together, these analyses demonstrate that sexual reproduction in *K. mangrovensis* involves recombination and can generate nonhaploid progeny with potential for self-fertility.

## Discussion

Our comparative genomic analysis revealed new insights into the evolution and structural dynamics of *Cryptococcus* and *Kwoniella MAT* loci, including distinct expansions of the *P/R* locus and additional independent *HD-P/R* fusion events, summarized in Fig 10 as a roadmap of key transitions.

### Multiple independent *P/R* locus expansions in *Cryptococcus* and *Kwoniella*

The archetypal basidiomycete *MAT* loci are thought to consist of a compact *P/R* region with one pheromone receptor and one pheromone gene, and an unlinked *HD* region containing two divergently transcribed homeodomain transcription

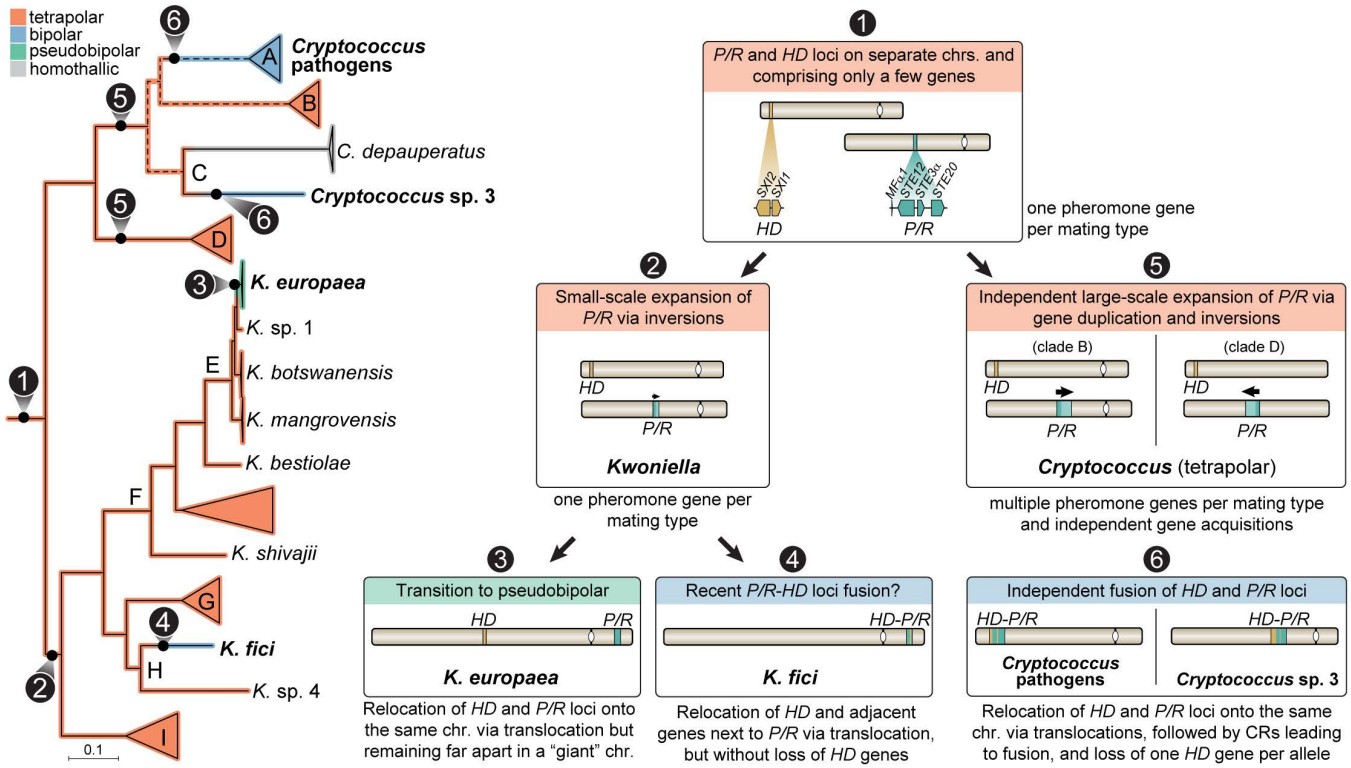

**Fig 10. Summary of the evolutionary transitions in *MAT* locus organization across *Kwoniella* and *Cryptococcus* species.** Schematic representation of key chromosomal rearrangements and evolutionary events underlying transitions from tetrapolar to bipolar and pseudobipolar mating configurations. The phylogenetic tree highlights inferred changes in *MAT* locus structure, with color-coded branches representing both extant and reconstructed *MAT* configurations. Dashed lines indicate unresolved phylogenetic relationships among clades A, B, and C. Insets summarize distinct evolutionary stages, including: the ancestral organization with unlinked and compact *P/R* and *HD* loci (1); small-scale expansion of the *P/R* locus in *Kwoniella* (2); two independent large-scale expansions of the *P/R* locus in *Cryptococcus* (5); relocation of *P/R* and *HD* loci onto the same chromosome in *Kwoniella* resulting in either pseudobipolar (3) or fused (4) *MAT* configurations; and independent *HD-P/R* fusion events with *HD* gene loss in *Cryptococcus* pathogens and *Cryptococcus* sp. 3 (6). These transitions illustrate the diverse pathways by which chromosomal rearrangements (CRs) have shaped *MAT* locus architecture in these fungal lineages.

factors (*HD1* and *HD2*). This configuration is retained in some lineages, such as in *Malassezia* species [20], but in most basidiomycetes the *P/R* locus has undergone expansions and structural changes, while the *HD* locus remains comparatively conserved (except in some rust fungi [109] and in mushroom-forming species with *HD* gene duplications [9,28]). For example, in *Ustilago maydis*, allele-specific differences in *P/R* locus size arise from two additional genes (*lga2* and *rga2*) in the *a2* allele that direct mitochondrial uniparental DNA inheritance [110,111]. Similarly, several *Microbotryomycete*s yeasts (*Pucciniomycotina*) harbor expanded *P/R* loci [26,112,113], as do tetrapolar *Cryptococcus* species [35,47,78]. Recent fully phased, chromosome-scale assemblies in rust fungi also uncovered multiple pheromone genes and *STE3* receptor duplications on single haplotypes [109,114], though the full structure of the *P/R* locus in these fungi remains incompletely resolved.

Our analyses across tetrapolar *Cryptococcus* and *Kwoniella* support independent, lineage-specific *P/R* locus expansions. The inferred ancestral *P/R* locus likely comprised a conserved four-gene core (*MF*, *STE3*, *STE12*, and *STE20*) (Fig 10, event 1). In *Kwoniella*, this region has remained relatively compact, with 9 core conserved genes (Fig 10, event 2), though lineage-specific modifications were detected, including mating-type-specific changes (e.g., *STE12* truncation in *a2* alleles) and small-scale expansions through the inclusion of genes likely entrapped by local inversions. We note that these

additions more likely reflect neutral consequences of inversions expanding recombination-suppressed regions, rather than adaptive recruitment, consistent with stepwise extension models of sex chromosome and *MAT* locus evolution [39,115]. In *K. ovata*, *K. endophytica*, and *K. dendrophila*, mosaic *P/R* configurations are better explained by past intra-locus recombination. This may have transiently enabled self-compatibility at the *P/R* locus, but the absence of self-filamentation in extant strains suggests a reversion to heterothallism. A full transition to homothallism would also require compatibility at the *HD* locus, either through mutations enabling pairing between otherwise nonself-compatible *HD1/HD2* alleles or through bypass of *HD* function, as seen in *C. depauperatus* [23].

In contrast to *Kwoniella*, tetrapolar *Cryptococcus* species from clades B and D exhibit pronounced *P/R* locus expansion, but with limited gene-content overlap, indicating that these expansions occurred independently after divergence from a common ancestor (Fig 10, event 5). A distinctive feature of the *Cryptococcus P/R* locus is the duplication of pheromone (*MF*) genes, with each mating type typically carrying multiple copies arranged in divergent orientations. Similar patterns occur in *Microbotryomycetes* yeasts [26,112,113,116,117], and have also been reported in rust fungi [109,114]. However, in rusts, some of the duplicated *MF* genes encode distinct pheromones, possibly reflecting adaptation to different receptor specificities, whereas in *Cryptococcus* and *Microbotryomycetes* the *MF* copies encode identical mature peptides, suggesting functional redundancy rather than diversification.

Experimental evidence in *C. deneoformans* supports a dosage-based model, in which all three *MFα* gene copies are expressed and contribute additively to mating efficiency and filamentation [118]. Triple mutants show severe mating defects, while overexpression enhances differentiation, indicating that pheromone dosage promotes mating responsiveness [118]. Although direct functional evidence in other *Cryptococcus* species is lacking, these duplications may similarly enhance partner recognition. This interpretation aligns with sexual selection frameworks in fungi, where pheromone production can act as a costly, honest signal of mating quality (the handicap principle), which posits that only high-quality individuals can afford the cost of producing strong signals and are therefore more likely to be chosen as mates [119,120]. Studies in model yeasts have shown preferential mating toward high pheromone producers [121,122], but equivalent tests in basidiomycetes are limited, and the ecological context for such selection in *Cryptococcus* remains unclear. Why pheromone duplications recur in *Cryptococcus* but not in *Kwoniella* remains uncertain. Their absence in *Kwoniella* may reflect lineage-specific constraints or ecological differences—for instance, *Kwoniella* species produce embedded rather than aerial spores, which could reduce the need for long-range mate signaling—although the lack of ecological and population-level data prevents firm conclusions.

Notably, *MF* gene duplications can also introduce structural instability by facilitating the formation of inverted repeats, which can promote inversion loops and allele-specific structural rearrangements, potentially expanding recombination-suppressed regions between *P/R* alleles. One way to mitigate this instability is by clustering *MF* gene pairs more closely, thereby restricting the extent of rearrangements. Consistent with this, we observed reduced structural variation in *C. neoformans MATα* strains and in the *C. gattii* species complex, where the two *MF* gene pairs are positioned in close proximity, separated by only two genes. In contrast, *MATa* strains exhibit a more dispersed arrangement of *MF* genes, which appears to contribute to greater structural diversity across strains, including the *C. neoformans* VNI-specific rearrangements that arose following introgression of the *MATa* allele from the VNBI lineage. Once established, multiple *MF* gene copies may be maintained as templates for intrallelic repair via gene conversion, a nonreciprocal exchange that can occur during meiosis between homologous sequences but also at lower frequencies between sister chromatids, or within duplicated regions on a single chromosome during mitosis [123,124]. If confirmed, this mechanism would resemble the role of palindromic sequences in preserving gene integrity on the human Y chromosome [125–127]. Introgression events, such as the transfer of *MATa* alleles between *C. neoformans* lineages, may further buffer against degeneration by resetting mutational load [128], thereby helping maintain complex *MAT* configurations.

Integrated with these structural features, our analyses revealed that the *P/R* locus in clades B and D *Cryptococcus* species has consistently lower GC% relative to genome-wide averages and that this depletion is best explained by reduced

gBGC in recombination-suppressed regions rather than by relaxed selection on codon usage. gBGC normally favors the fixation of G/C alleles in regions of frequent recombination, and its reduction allows the AT-biased mutational spectrum to dominate over time, driving local base composition shifts. Similar GC depletion has been reported in the *Trichosporonales MAT* locus [18]. Interestingly, this trend is not observed in the *MAT* locus of pathogenic *Cryptococcus*, suggesting that genomic GC background also influences the outcome. In high-GC genomes (e.g., clades B and D), recombination suppression accentuates AT accumulation, whereas in lower-GC genomes (pathogenic *Cryptococcus*; [49]), the absence of a shift suggests they may already be near equilibrium under mutation pressure, masking localized effects.

## Independent transitions to bipolar and pseudobipolar via chromosomal translocations and inversions

Beyond the previously documented tetrapolar-to-bipolar transition in pathogenic *Cryptococcus* lineages (clade A), we identified three additional transitions: two leading to bipolarity (*Cryptococcus* sp. 3 and *K. fici*; Fig 10, events 6 and 4), and one resulting in a pseudobipolar arrangement (*K. europaea*; Fig 10, event 3). Our findings suggest that the transition to bipolarity in *Cryptococcus* sp. 3 likely occurred independently of that in pathogenic species. First, fusion of the *P/R* and *HD* loci in *Cryptococcus* sp. 3 was unrelated to the reduction from 14 to 13 chromosomes and instead originated from a translocation followed by an inversion. Second, the *MAT* locus in this species retains only *SXI2* (*HD2*) linked to the *STE3α*, unlike in pathogenic species where *SXI2* is associated with *STE3***a**. Third, phylogenetic analyses revealed that many genes within the *Cryptococcus* sp. 3 *MAT* locus do not group with **a** or α allele-specific clusters from pathogenic species. If the fusion had predated the divergence of these clades, trans-specific polymorphism would be expected, but this pattern was not observed, which is more consistent with an independent origin.

However, whether the initial translocation event that brought the *P/R* and *HD* loci onto the same chromosome occurred independently in *Cryptococcus* sp. 3 or in a common ancestor of clades A and C, remains unresolved. This uncertainty stems from three main factors. First, phylogenetic ties among clades A (bipolar), B (tetrapolar), and C (which includes *Cryptococcus* sp. 3 and *C. depauperatus*) are still ambiguous. Second, although our synteny analyses indicate that the translocation breakpoints at the origin of the physical linkage between *MAT* loci differ between *Cryptococcus* sp. 3 and pathogenic species (supporting independent events), these differences might also reflect subsequent rearrangements that accumulated along the two lineages. Third, we have previously reported that *C. depauperatus* also shows a partial merging of *HD*- and *P/R*-derived genes into the same genomic region, but with more extensive gene reshuffling and *HD* gene loss likely associated with a transition to homothallism [23]. Further taxon sampling, combined with phylogenomic approaches tailored to deep divergences, will be required to conclusively resolve these relationships and clarify the broader implications of these transitions.

The tetrapolar-to-bipolar transition in *Cryptococcus* pathogens has been hypothesized to result from a chromosomal translocation mediated by ectopic recombination between repetitive elements in the centromeric regions of the *P/R*- and *HD*-containing chromosomes [35]. This model had support from several observations: (i) *Cryptococcus* centromeres are enriched with long-terminal-repeat retrotransposons (LTRs) from the Ty3 and Ty1 superfamilies, which are shared among different centromeres and may facilitate recombination [23,47,49,129,130]; (ii) chromosomal arm exchanges occurred during the divergence of *Cryptococcus* lineages from their common ancestor [35,108]; and (iii) the repair of experimentally induced double-strand breaks at centromere-specific transposons in *C. neoformans* frequently results in chromosomal translocations [108]. However, our broader comparisons indicate that the centromere of the *C. neoformans MAT* chromosome, along with extended flanking regions, is conserved across most *Cryptococcus* species and in *K. shandongensis*, suggesting an ancestral organization.

Indeed, our analyses suggest that translocations outside centromeric regions may have played a more prominent role in repositioning the *P/R* and *HD* loci onto the same chromosome (Fig 7). Such noncentromeric translocations also appear to underlie the transition to bipolarity in several *Ustilaginales* species, including *Sporisorium scitamineum* [30] and in the common ancestor of *Ustilago hordei*, *U. nuda*, and *U. bromivora* [31]. Noncentromeric translocation events are also

common across other genomic regions when comparing genomes of different *Cryptococcus* species [49] and even among conspecific strains of *C. neoformans* [101,131]. Whether such translocations are influenced by TEs has not yet been systematically analyzed, but studies in *Saccharomyces cerevisiae* have shown they can occur via homologous recombination between retrotransposons on different chromosomes [132]. Given the abundance of TEs in *Cryptococcus* genomes [49], they may likewise have facilitated the chromosomal relocation of *MAT* loci and the subsequent transition to bipolarity. Our findings offer an alternative to models invoking intercentromeric recombination as the primary event initiating the tetrapolar-to-bipolar transition in *Cryptococcus* pathogens.

Genetically linked *P/R-HD* loci can also arise from the fusion of entire ancestral *P/R* and *HD* chromosomes, as reported in *Microbotryum violaceum* on *Silene paradoxa* [27]. In *Kwoniella*, repeated and independent chromosome fusions have drastically reduced chromosome numbers, with some extant species retaining only 3 (clade E) or 5 (clades G and H) chromosomes from a 14-chromosome ancestral karyotype [49]. Yet, aside from the bipolar arrangement in *K. fici* (clade H) and the pseudobipolar arrangement in *K. europaea* (clade E), no other instances of *P/R-HD* linkage or colocation were detected in *Kwoniella*, and even in these species the transition reflects chromosomal translocations rather than whole-chromosome fusions.

In *K. fici*, the transition to fused *MAT* loci did not result in loss of *SXI1* (*HD1*) or *SXI2* (*HD2*), possibly suggesting a relatively recent fusion where insufficient time has passed for one *HD* gene to decay in each mating type, as observed in *Cryptococcus* sp. 3, *Cryptococcus* pathogens [133], and *Trichosporonales* [18]. Other known instances of *HD* gene loss in bipolar species, albeit distinct, include: (i) *Microbotryum violaceum* on *Silene caroliniana*, where *HD1* is absent in *a2* strains, while *a1* retains both *HD* genes [27]; (ii) *Microbotryum superbum*, where loss of *HD* function in mating compatibility may have relaxed constraints on *HD2* maintenance in *a1* strains, leading to its disruption while *a2* strains retain both *HD* genes [117]; and (iii) xerotolerant *Wallemia* species, where *HD1* is retained in *MAT a1*, while the opposite mating type appears to lack any *HD* gene [134]. Because *HD* compatibility is no longer an independent determinant of mating success in bipolar species, maintaining both *HD* genes in each *MAT* allele may become dispensable, with loss potentially reflecting genomic streamlining once *MAT* loci are fused and recombination suppressed. Yet, as in *K. fici*, the retention of both *HD* genes in each allele remains the predominant organization in other bipolar basidiomycetes with fused *MAT* loci, including several *Microbotryum* [27], *Sporisorium* [30], and *Ustilago* species [31,135,136], possibly reflecting differences in mating strategies or genomic constraints. For instance, if both *HD* genes have additional roles beyond mating-type determination that do not require heterodimerization, their retention could be selectively maintained, although this possibility remains largely unexplored in basidiomycetes.

In contrast to *K. fici*, where the *MAT* loci are fused, *K. europaea* has the two loci ~8.5 Mb apart on the same chromosome. This distance makes tight genetic linkage unlikely, though extended recombination suppression over megabase-long spans has been reported in *Neurospora tetrasperma* and *Microbotryum* [39,137]. Because crosses in *K. europaea* have not yielded sexual reproduction [66], it remains unclear whether this configuration represents an incipient step toward bipolarity or a neutral byproduct of chromosomal rearrangements. If recombination persists, this system would functionally resemble tetrapolarity, as the two loci would still reassort in most meioses, making *MAT* loci colocation unlikely to meaningfully increase sibling compatibility. In this case, the arrangement may simply reflect a stochastic translocation rather than a selectively maintained feature.

Together, these observations reinforce the view that chromosomal rearrangements provide the mechanistic basis for tetrapolar-to-bipolar transitions but do not inherently drive shifts in breeding systems. Instead, whether a lineage remains tetrapolar or transitions to bipolarity likely depends on ecological and evolutionary pressures such as opportunities for outcrossing versus inbreeding, the frequency of sibling or clonal mating after colonization, dispersal limitations, and the long-term stability of chromosomal rearrangements that maintain linkage between the loci. Accordingly, species with linked *MAT* loci are often associated with animals or plants as commensals or pathogens, where linkage facilitates mating among siblings and provides an advantage during host colonization [9,28,81,138,139]. A similar scenario has been

proposed for the *Trichosporonales*, where *MAT* locus fusion may have originated in a host-associated ancestor and persisted in saprobic descendants [18]. Our results, however, show that bipolarity is not restricted to pathogens: *Cryptococcus* sp. 3 (isolated from bark beetles) and *Kwoniella fici* (isolated from fig) harbor fused *MAT* loci with no evidence of pathogenicity to date. This suggests that restricted partner availability—whether in insect-associated microhabitats or free-living clonal populations—may favor *MAT* loci fusion independently of pathogenicity. Once established, bipolar systems may be difficult to revert, and their stability could in turn facilitate the persistence or emergence of pathogenic lifestyles by ensuring mating success under the constrained conditions typical of infection.

## Outlook

Our characterization of *MAT* loci gene content and organization allowed us to infer putative breeding systems (heterothallism versus homothallism) across species. However, determining how often sexual reproduction and recombination occur in nature, and whether mating involves primarily outcrossing or inbreeding, will require population-level analyses such as linkage disequilibrium decay [140]. Unlike basidiomycete macrofungi, where sexual structures can be directly observed in the field, or smut fungi, where disease symptoms visibly reflect the production of sexual spores, most *Tremellomycetes* are inconspicuous environmental isolates or parasites of lichens and other fungi, complicating the isolation of compatible mating partners. Their cryptic nature has led to many species being described from a single isolate [48], hindering mating assays. Progress will require expanded taxon and population-level sampling [48,50], particularly for nonpathogenic species, to clarify reproductive strategies, transitions in mating systems, and potentially capture changes in *MAT* locus organization as they occur. Uncovering the hidden diversity of *Cryptococcus* species—particularly in their likely African origin [50]—will also be crucial for linking ecology, speciation, and adaptation with the evolutionary forces shaping *MAT* locus evolution and reproductive transitions, with possible implications for pathogenic emergence.

## Materials and methods

### Strains and media

Strains were routinely grown on YPD medium (10 g/L yeast extract, 20 g/L Bacto Peptone, 20 g/L dextrose, and 20 g/L agar) unless indicated otherwise. *C. neoformans* and *C. deneoformans* strains were incubated at 30 °C, while other *Cryptococcus* and *Kwoniella* species strains were grown at room temperature (20–23 °C). A complete list of strains is provided in S1 Appendix. Strain NCYC1536 was obtained from the National Collection of Yeast Cultures (Norwich, UK), and strain CMW60451 was isolated in October 2022 from a bark beetle (*Lanurgus* sp.) infesting twigs of *Widdringtonia cedarbergensis* in the Cederberg Mountains, South Africa.

### Genomic DNA extraction

High-molecular-weight DNA was extracted with a cetyltrimethylammonium bromide extraction as previously described [108], minimizing DNA shearing during sample preparation. DNA quality was evaluated by determining the A260/A280 and A260/A230 ratios on NanoDrop spectrophotometer (Thermo). Integrity and fragment size were analyzed using clamped homogeneous electric field electrophoresis, and gDNA for short-read whole-genome sequencing (Illumina) was extracted with a phenol:chloroform-based protocol, both as previously described [49]. DNA concentration was measured with Qubit dsDNA Assay Kits (Invitrogen) on the Qubit fluorometer.

### Genome sequencing

Whole-genome sequencing was performed with PacBio, Nanopore, and Illumina technologies. For *K. heveanensis* BCC8398 and *K. mangrovensis* CBS10435, Illumina sequencing at the Broad Institute Genomics Platform

utilized "fragment" and "jumping" libraries, prepared and sequenced as previously described [49]. All other Illumina sequencing was conducted at the Duke University Sequencing and Genomic Technologies (DUSGT) Core, with libraries prepared with Kapa HyperPlus kit and sequenced as paired-end 2 × 150 bp reads on various Illumina platforms. For PacBio sequencing, 15–20 kb insert-size libraries were prepared and run on a PacBio RS II or Sequel (2.0 chemistry) system at the DUSGT. Nanopore sequencing was carried out in-house. Single-strand libraries were prepared with either the SQK-LSK108 or SQK-LSK110 kit, whereas up to three DNA samples were barcoded with the SQK-LSK109 and EXP-NBD103/EXP-NBD104, or SQK-NBD114.24 kits. Libraries were prepared following the manufacturer's protocols, either individually or pooled, and sequenced on R9 (FLO-MN106) or R10 (FLO-MIN114) flow cells for 48–72 h at default voltage on a MinION Mk1B or Mk1C system. The MinION software version current at the time of each run was applied. Further details on sequencing platforms, basecalling, and demultiplexing are provided in S1 Appendix.

## Genome assembly

Genomes were assembled with Canu [141] using Nanopore or PacBio sequencing data and default parameters. Assembly accuracy was improved through initial error correction with Medaka (https://github.com/nanoporetech/medaka) for Nanopore-based assemblies, followed by up to five rounds of iterative polishing with Pilon v1.22 [142] (--fix all) using Illumina reads aligned to the first pass-polished assembly with BWA-MEM v0.7.17-r1188 [143]. Pilon iterations were stopped early if no additional changes were detected, based on the absence of reported corrections in each round. Contigs containing only rDNA sequences, detected by Barrnap (https://github.com/tseemann/barrnap) (--kingdom euk), or those classified as mitochondrial DNA, were excluded from the final nuclear genome assembly. To confirm assembly completeness and evaluate telomeric regions, Nanopore/PacBio and Illumina reads were realigned to the Canu-corrected assembly using minimap2 v2.9-r720 [144] and BWA-MEM, respectively, and read coverage profiles were examined in the Integrative Genomics Viewer [145]. Draft assemblies for *K. europaea* PYCC6162 and *K. botswanensis* CBS12717 were generated with SPAdes v3.15.3 using default settings, while assemblies for *K. heveanensis* BCC8398 and *K. mangrovensis* CBS10435 were constructed with Allpaths [146]. Genome assemblies and raw sequencing data were deposited in DDBJ/EMBL/GenBank under the BioProject numbers listed in S1 Appendix, which also provides details on assembly parameters for each genome.

## Gene prediction and annotation

Gene models were predicted using either BRAKER2 v2.1.5 [147] or Funannotate v1.8.9 (https://github.com/nextgenusfs/funannotate), as previously described [20,23,49]. Prior to gene prediction, assemblies were masked with RepeatMasker v4.1.4 in sensitive mode, applying a custom library that combined *Cryptococcus–Kwoniella* repeat families generated with RepeatModeler2 v2.0.4 [49], centromere-associated TEs, and curated *Cryptococcus* elements compiled from published sources [148] and represented in Dfam. Manual inspection and correction were performed only for mating-type gene models, guided by ortholog alignments, and included adjustments to exon–intron boundaries or addition of small ORFs missed by automated pipelines (e.g., *MF* pheromone genes). These curated annotations were incorporated into the final GFF files and are part of the genome submissions deposited in NCBI. Functional annotation was added to the final gene models using the Funannotate "annotate" module, which integrated data for PFAM and InterPro domains, Gene Ontology terms, fungal transcription factors, Cluster of Orthologous Genes (COGs), secondary metabolites, Carbohydrate-Active Enzymes (CAZYmes), secreted proteins, proteases (MEROPS), and Benchmarking Universal Single-Copy Orthologs (BUSCO) groups. InterPro domain data, COG annotations, and secondary metabolite predictions were generated with InterProScan v5.55-88.0, eggNOGmapper v.2.1.7 (eggNOG DB version: 5.0.2) [149], and AntiSMASH v6.1.0 [150], and then passed to Funannotate annotate with the options --iprscan, --eggnog, and --antismash. Specific parameters are provided in S1 Appendix.

## Ortholog identification and sequence alignment

To construct the phylogenomic data matrix, single-copy orthologs (SC-OGs) were identified across *Cryptococcus* and *Kwoniella* species and three outgroups (*Tremella mesenterica* ATCC28783, GCA_004117975.1; *Saitozyma podzolica* DSM27192, GCA_003942215.1; and *Bullera alba* JCM2954, GCA_001600095.1) using OrthoFinder v3.0.1b1 with options: `-M msa -S diamond_ultra_sens -I 1.5 -M msa -A mafft -T fasttree -t 48 -a 6`. This identified 3,086 SC-OGs shared across species. Their amino acid sequences were individually aligned with MAFFT v7.310 [151] (`--localpair --maxiterate 1000`) and trimmed with TrimAl v1.4.rev22 [152] (`-gappyout -keepheader`). The same approach was applied to construct the phylogenomic data matrix for *C. neoformans* strains representing different VN groups, with three *C. deneoformans* outgroups, yielding 5,439 shared SC-OGs, which were then aligned and trimmed as above.

## Species phylogeny, estimation of topological support, and gene genealogies

The updated phylogeny of *Cryptococcus* and *Kwoniella* species (Figs 1A and S1) was inferred using two complementary strategies: (i) concatenation-based partitioned ML phylogeny reconstruction in IQ-TREE v2.1.6 [153], and (ii) gene-based coalescence analysis in ASTRAL v5.7.8 [154]. For the concatenation approach, amino acid alignments of 3,086 SC-OGs were combined into a partitioned supermatrix (52 taxa, 3,086 partitions, 1,686,691 sites) using the "-p" option in IQ-TREE. The edge-linked proportional partition model [155] was applied to account for differences in evolutionary rates across partitions, and the best-fit substitution model for each partition was identified with ModelFinder [156] under the Bayesian information criterion. The ML tree was inferred with the parameters `--seed 54321 -m MFP -msub nuclear -B 1000 -alrt 1000 -T 14`, incorporating 1,000 ultrafast bootstrap (UFBoot) replicates [157] and Shimodaira–Hasegawa approximate likelihood ratio tests (SH-aLRT) for branch support assessment. For the coalescence-based approach, ML gene trees were constructed for each SC-OG alignment using IQ-TREE's "-S" option, which performs both model selection and tree inference for individual alignments. These trees were then input to ASTRAL under default settings to reconstruct the species phylogeny. Quartet support values were computed with the `-t 2` option, providing quartet support metrics for the primary topology (q1) and alternative topologies (q2, q3), and local posterior probability (LPP) support. Genealogical concordance and topological support for branches in the concatenated ML tree were evaluated with gene concordance factor (gCF) and site concordance factor (sCF) metrics in IQ-TREE. This analysis utilized both the best-scoring concatenated ML tree (concat.treefile) and the set of gene trees (loci.treefile), with options `-t --gcf -p --scf 100`. The concatenation-based approach was also applied to resolve relationships among *C. neoformans* VN lineages (Fig 4A). For gene genealogies, amino acid sequences of individual genes of interest were extracted from OGs identified by OrthoFinder, manually inspected, and reannotated as needed. Alignment, trimming, and ML reconstruction were performed as described above. Trees were visualized and annotated with iTOL v7. Detailed model parameters are provided in the tree files accessible at https://doi.org/10.5281/zenodo.16987438.

## Synteny analyses, *MAT* loci delineation, and centromere identification

Conserved synteny blocks in pairwise genome comparisons were identified with SynChro [105], using a delta parameter of three for high stringency. Comparisons in Figs 6A, 8A and S9A employed the *K. shandongensis* genome as reference. Detailed linear synteny plots of chromosomes and specific genomic regions, including *MAT* loci, were generated with EasyFig v2.2.2 [158] using BLASTN. Centromeres positions were determined in silico as previously described [49], combining detection of centromere-associated LTR elements and synteny analysis. Centromere lengths were defined as the intergenic regions between flanking centromeric genes (S1 Appendix). *MAT* loci were initially identified by BLAST searches, with *C. neoformans*-derived *MAT* genes and their flanking proteins as queries. The *HD* loci were defined as the regions spanning the *HD1* and *HD2* genes. The *P/R* loci were delineated based on structural comparisons between mating types. For species with available strains of opposite mating types, the *P/R* locus was defined as the region where

synteny between the mating types is disrupted, with boundaries set at the points where synteny is restored. For species with only a single strain, the *P/R* locus length was inferred from the distance between *P/R*-flanking genes, as determined from comparisons involving opposite mating types.

To infer the origin of structural rearrangements observed in the *P/R* locus in *Kwoniella* (S5A Fig), we manually inspected the orientation and synteny of *P/R*-associated genes across mating types and species. Rearrangement breakpoints were approximated by identifying changes in gene order and orientation between alleles, supported by the presence of gene fragments (e.g., partial *NDC80*, *STE12*, *BSP3*) and conserved synteny in the opposite mating type. These rearrangements were reconstructed by inverting or repositioning affected gene blocks in the more derived *a2* allele to match the putative ancestral configuration in the more conserved *a1* allele.

Statistical analyses of *MAT* loci lengths across species, clades, or mating types were conducted with Python3 with Pandas, Seaborn, Matplotlib, and SciPy libraries. Differences between the two groups were assessed using the two-sided Mann–Whitney *U* test. Scripts and raw data are available at https://doi.org/10.5281/zenodo.16987438. Plots were refined for clarity by adjusting color schemes, labels, and adding features in Adobe Illustrator.

### Analysis of gene content in *MAT* loci, frequency across species, and gene essentiality classification

A curated gene presence/absence matrix was constructed after delineating *MAT* loci across species. Genes were categorized as "present" (1), "absent" (0), "unclear" (?), or "pseudogene" (pseudo). Unclear cases indicated instances where gene presence could not be conclusively determined due to uncertain locus boundaries. For frequency calculations, pseudogenes were treated as present and unclear cases as absent. The presence of either *SXI1* or *SXI2* was scored as present and combined for frequency estimates. Information on gene essentiality was inferred from *C. neoformans* H99, based on experimentally validated studies [34,159,160] or predictions from a high-throughput transposon mutagenesis (TN-seq) study [92]. Gene essentiality in other species was extrapolated solely from H99, and genes without H99 orthologs were categorized as "unknown" due to the lack of supporting data. A custom R script (`0_calculate_gene_pres-ence_frequency.R`) analyzed and plotted gene presence frequencies. The matrix was also visualized as a heatmap (`1_plot_gene_matrix_based_on_frequency.R`), with genes ordered by frequency. *SXI1* and *SXI2* are shown as separate columns in the heatmap, but their ranking reflects their combined frequency. Analyses and visualizations were conducted in R with dplyr, ggplot2, ComplexHeatmap, and circlize packages. Gene presence/absence was further analyzed with a Python script (`2_compare_MAT_gene_content_across_clades.py`), which calculated clade-specific and pairwise presence, identified genes shared across specified clades, and highlighted clade-specific genes. Input matrices are provided in S2 and S3 Appendices and scripts are available at https://doi.org/10.5281/zenodo.16987438.

### Base composition and codon-bias analyses

Genome-wide GC content and deviations from the mean GC% were analyzed with a custom Python script (`00_gc_con-tent_analysis_and_plots.py`), employing a nonoverlapping sliding window (1 kb for genome scans; 0.25 kb for zoomed-in views). Regions of interest (e.g., *P/R* loci) were highlighted from BED file coordinates. Plots were further refined in Adobe Illustrator for producing Fig 2. To assess whether the *P/R* locus exhibited lower GC content than genome-wide averages, a second script (`01_gc_content_analysis.py`) compared *P/R* locus GC% to 1,000 randomly sampled genome-wide regions of equal size (excluding the *P/R* locus). A one-sample *t* test assessed whether the *P/R* locus GC content differed from the sampled mean, while a permutation test (10,000 iterations) generated a null distribution to calculate *P*-values and *Z*-scores. The analysis was automated with a shell script (`01a_run_script_vs_wg.sh`), consolidating results into a summary table (S4 Appendix).

To assess whether GC depletion within the *P/R* locus reflects reduced gBGC or relaxed selection on codon usage, we performed a four-step Python3-based analysis (NumPy, pandas, SciPy, Biopython): (i) neutral noncoding GC comparison, (ii) coding third-position composition (AT3 versus GC3), (iii) codon-bias indices (Fop, Nc, ENC, ΔENC), and (iv) CAI

analyses controlling for GC3 and CDS length (details in S1 Text). As preparation for CAI-related analyses, species-specific CAI reference sets were generated with a custom script (`02_build_cai_reference.py`) in "rp_only" mode, selecting ribosomal-protein genes as proxies for highly expressed, codon-optimized, genes [161,162], since RNA-seq data for these species was unavailable. The script counted codons and generated a CAI weight table in which, within each synonymous family, relative adaptiveness weights ($w$) were normalized so that the most frequent codon had $w = 1$ and others $0 < w \le 1$. Per-gene CAI values were then computed with `03_compute_cai_per_gene.py`, which reads each species' CAI weight table and CDS annotations and returns gene-level CAI (geometric mean of w across codons; start/stop excluded; nonACGT ignored; Met/Trp treated as neutral set to $w = 1$).

To test for GC depletion in the *P/R* region, noncoding GC within *P/R* (intergenic plus introns) was compared to same-chromosome, length-matched nulls with script `04_neutral_non_coding_GC.py` using 10,000 *P/R*-excluded windows (`--n_null 10000 --no_overlap --min_gap 1000`). The script reports ΔGC (*gc_intergenic − null_mean_intergenic*), a standardized $z$, and a one-sided empirical permutation $p$-value (PR ≤ null).

Coding third-position composition (AT3 versus GC3) was analyzed with script `05_coding_composition_and_codon_bias.py` by counting AT- versus GC-ending third positions inside *P/R* and background regions (same chromosome, outside *P/R*) and testing the 2 × 2 table by $\chi^2$ with Fisher's exact fallback. Using the same script and the CAI weights above, codon-bias indices were calculate: Fop (frequency of optimal codons based on the CAI reference); Nc (effective number of codons, representing the observed codon usage bias in a gene); ENC (the expected Nc based solely on the gene's GC3, calculated from the GC3–Nc relationship described by Wright (1990) [163]); and ΔENC = Nc − ENC (representing codon bias beyond that expected from GC3 composition alone). To improve the stability of Nc estimates, particularly for short CDS, we applied a small pseudocount (0.5 per codon), excluded very small synonymous families, bounded intermediate calculations to avoid division by near-zero values, and constrained Nc values to the theoretical range of 20 (maximum bias) to 61 (no bias) before computing ΔENC. *P/R* versus background contrasts were evaluated as median differences using Mann–Whitney tests with 10,000 permutations.

Finally, CAI differences independent of nucleotide composition were tested using three complementary analyses implemented in script `06_cai_vs_gc3_controls.py` (detailed in S1 Text), using the following parameters: `--n_perm 10000`, `--gc3_tol 0.03`, `--len_tol 0.10`, and `--k_match 5`. Briefly, we (i) fit CAI ~ inside_pr + Δ(exon-intron GC) + log(length_cds) to all genes using ordinary-least-squares (OLS) with heteroskedasticity-consistent (HC1) robust standard errors; inside_pr = 1 for CDS fully within the *P/R* window and 0 for same-chromosome genes outside (boundary overlaps excluded); (ii) we trained CAI ~ Δ(exon-intron GC) + log(length_cds) on background genes only, computed residuals for all genes, and tested median(PR residual) − median(BG residual) with a one-sided permutation test (10,000 shuffles; alternative PR < BG); and (iii) we matched each *P/R* locus gene to same-chromosome controls within ±0.03 GC3 and ±10% CDS length (dropping unmatched *P/R* genes), computed *CAI_match_median = median[CAI_PR − mean(CAI_BG matches)]*, and used a sign-flip permutation test (10,000 flips; alternative PR < BG). Fop was analyzed identically to step (iii) with the same matching tolerances and permutation settings. All scripts are available at https://doi.org/10.5281/zenodo.16987438, and per-species summaries and statistical tests are given in S4 Appendix.

### Variant analysis and SNP distribution in *C. neoformans MAT*a strains

Multi-genome variant analysis was performed with the Snippy pipeline v4.6.0 (https://github.com/tseemann/snippy) using the newly assembled genome of *C. neoformans* Ftc555-1 (VNBI) as the reference, with parameters: `--cpus 10, --unmapped, --mincov 10, and --minfrac 0.9`. Paired-end reads were obtained from the NCBI SRA database (S1 Appendix). Three datasets were independently used: (i) the entire genome, (ii) only the *MAT* locus region (chr_5:153,773-277,230), and (iii) only chr. 5. The *MAT* locus and chr. 5 were extracted from the reference genome to generate region-specific GenBank files for Snippy to identify SNPs and construct region-specific phylogenies. Visualization of SNV distributions was based on the whole-genome dataset and included both chromosome-wide and zoomed-in

views of specific regions. A custom Python script processed merged SNV datasets, computed total SNP counts for each strain, and generated gradient color-coded SNP density plots. Final composite figures were refined using Adobe Illustrator. Core SNP alignments (core.aln) from each dataset were used for phylogenetic reconstruction with IQ-TREE2, incorporating 10,000 UFBoot replicates and SH-aLRT tests for branch support. Scripts for SNV analysis and visualization are available at https://doi.org/10.5281/zenodo.16987438.

## Chromosome composition analysis of *K. mangrovensis* progeny

To characterize chromosome composition in *K. mangrovensis* progeny, a combined nuclear reference genome was constructed from assemblies of the two parental strains, CBS8507 and CBS10435. For CBS8507, the original assembly consisted of four contigs, two of which represented the same chromosome fragmented at the rDNA array. To streamline downstream analyses and facilitate interpretation, the two contigs were merged into a single scaffold: orientation was first established from rDNA gene direction; the corresponding ends were aligned and trimmed; and a 5-N gap was inserted to join them, yielding a finalized three-chromosome assembly. For CBS10435, the draft assembly of 37 contigs was reordered and reoriented to match the CBS8507 3-chromosome assembly using D-GENIES [164], which employs minimap2 for genome alignment. Raw Illumina paired-end reads from selected progeny were then mapped to a combined reference with the sppIDer pipeline [165], which sequentially mapped reads, applied stringent quality filtering (MQ>3), and generated depth-of-coverage plots. Chromosome number and ploidy were estimated by integrating spider coverage data with flow cytometry results.

## Mating assays, phenotyping on mating-inducing media, and microscopy

Mating assays with *K. mangrovensis* strains CBS8507 and CBS10435 were performed as previously described [66,67]. Equal amounts of cells from each strain were mixed on V8 agar (10 g/L yeast extract, 20 g/L Bacto Peptone, 20 g/L dextrose, 20 g/L agar; pH 5) or corn meal agar (CMA; 15 g/L corn meal agar, 5 g/L agar) and incubated in the dark at room temperature (20–23 °C) for up to 1 month. Plates were regularly monitored for the development of mating structures. Random F1 progeny were recovered by microdissecting individual cells from cell clusters [79] embedded in or on the surface of the mating media (V8 and CMA) at the edges of mating patches. A total of 51 progeny were collected, grown axenically on YPD plates, and stored as glycerol stocks at −80 °C. Selected F1 progeny (MP11, MP15, MP16, MP18, MP49, MP50, MP51, MP54, and MP59) were assessed for hyphal formation and mating structures on V8 pH5 and CMA, compared to the parental cross (CBS8507 × CBS10432) and the solo culture of each parental strain. Potentially compatible strain pairs from *Cryptococcus* species lacking documented sexual reproduction (IND107 × CBS11718; CBS11687 × 7685027; and DSM108351 × NCYC1536) were tested for mating ability under conditions known to induce mating in *C. neoformans* and *C. deneoformans* (V8 media, dark, room temperature) for up to a month, with regular monitoring [79]. Solo cultures of each strain were also evaluated for their ability to reproduce sexually without a compatible partner. Similar assays were performed for strains of newly described species lacking known compatible partners. All mating assays were repeated three separate times to ensure consistent phenotypic observations. To assess hyphal growth, basidia, and spores, the edges of yeast colonies and mating patches were examined and photographed with a Zeiss Axio Scope.A1 microscope equipped with an Axiocam Color camera and ZEN Lite v3.4 software.

## Analysis of *K. mangrovensis* progeny by PCR-restriction fragment length polymorphism (RFLP) analysis

*Kwoniella mangrovensis* has only three chromosomes. To analyze the F1 progeny from the CBS8507 × CBS10432 crosses, six primer pairs (one per chromosomal arm) were designed for the initial screening. Primers were manually designed to produce distinct parental patterns after PCR product digestion with specific restriction enzymes. As a heterozygosity control, PCR products from a 1:1 DNA mixture of both parental strains were amplified for each primer pair,

followed by digestion with the corresponding restriction enzyme. Results and primer sequences are provided in S5 Appendix.

**Fluorescence-activated cell-sorting (FACS)**

FACS was performed as previously described [166] to determine ploidy of *K. mangrovensis* CBS8507, CBS10432, and nine selected progeny. Strains were grown overnight at room temperature (21–23 °C) on YPD medium, harvested, and washed with PBS. Cells were fixed in 2 ml of 70% ethanol at 4 °C overnight, washed with 1 ml of NS buffer (10 mM Tris-HCl pH 7.2, 250 mM sucrose, 1 mM EDTA pH 8.0, 1 mM MgCl$_2$, 0.1mM CaCl$_2$, 0.1 mM ZnCl$_2$, 0.4 mM phenylmethylsulfonyl fluoride, and 7 mM β-mercaptoethanol), and then stained with 5 μl of propidium iodide (0.5 mg/ml) in 180 μl NS buffer with 20 μl of RNase (10 mg/ml) at 4 °C overnight. Lastly, 50 μl of stained cells were diluted in 2 ml of 50 mM Tris-HCl pH 8.0 and sonicated for 1 min before analysis at the Duke Cancer Institute Flow Cytometry Shared Resource. Data were collected from 10,000 cells using the FL1 channel on a Becton-Dickinson FACScan and analyzed with FlowJo software. Strains JEC21 and XL143 served as haploid and diploid controls, respectively [167].

## Supporting information

**S1 Fig. *Cryptococcus* and *Kwoniella* species phylogenies.** Both trees were inferred using a dataset of 3,086 single-copy genes shared across all species and three outgroups (depicted in gray). **(A)** Phylogeny inferred using a concatenation-based approach, corresponding to the tree in Fig 1A, but with all branch support and concordance values displayed (SH-aLRT, UFBoot, gCF, and sCF). **(B)** Tree topology inferred using a coalescence-based approach (ASTRAL), with local posterior probability (LPP) values shown for each branch. Quartet values for the main topology (q1) and alternative topologies (q2 and q3) are included for branches where q1 < 0.4, reflecting some uncertainty considering the expected value of 0.33 for a hard polytomy.
(PDF)

**S2 Fig. Assessing sexual reproduction in *Cryptococcus* species.** All micrographs are from cultures incubated for 2 weeks in the dark at room temperature on the indicated media. For each panel, the left image shows the colony edge, and the right image shows a higher-magnification view of cell or hyphal morphology from the same culture. **(A)** *C. decagattii*: solo cultures of strains 7685027 (*MAT*α) and CBS11687 (*MAT***a**) produced only yeast cells (right panels). In contrast, the α × **a** cross generated hyphae at the colony margin, with basidia and a few elongated basidiospores (arrowheads). Long spore chains were not observed, possibly due to rapid spore discharge. **(B)** *C. tetragattii*: solo cultures of strains IND107 (*MAT*α) and CBS11718 (*MAT***a**) produced only yeast cells (right panels), and the α × **a** cross did not form hyphae or sexual structures, even after prolonged incubation (>1 month). **(C)** Solo cultures of *Cryptococcus* sp. 3 (CMW60451) and *C. depauperatus* (CBS7841) on YPD and V8 pH 5 media. *C. depauperatus* produced abundant hyphae on both media; however, basidia (arrowhead) and spore chains (brackets) were only observed under V8 conditions. In contrast, *Cryptococcus* sp. 3 grew primarily as yeast on YPD, occasionally forming pseudohyphae on V8 pH 5, but no sexual structures were detected even after extended incubation (>1 month). **(D)** *Cryptococcus* sp. 4: solo cultures of NCYC1536 (*a1b1*) and DSM108351 (*a2b2*) grew as yeast (right panels). The *a1b1* × *a2b2* cross produced limited hyphae (right panel, zoomed view), but these were incipient and nonrecurrent, and no sexual structures were observed even after >2 months. Scale bars: 200 μm (colony edge images); 10 μm (right panels), except for panel D bottom right = 25 μm.
(TIF)

**S3 Fig. Structure and genomic context of the *HD* mating-type locus in tetrapolar *Cryptococcus* and *Kwoniella* species.** This supplementary figure spans five pages and contains a total of 15 panels (labeled A–O), with three consecutive panels per page. For each group of three panels, the top-left panel shows a synteny view of the full chromosomes, highlighting the chromosomal location of the *HD* locus; the bottom panel provides a zoomed-in synteny view (~200 kb)

centered on the *HD* locus; and the top-right panel displays a simplified phylogenetic tree, providing contextual information for the species included in the synteny analyses. For cross-referencing, *HD*-associated genes in the zoomed-in panel are colored gold if the corresponding ortholog in *Cryptococcus* pathogens is located within the *MAT* locus or shown in a darker shade when positioned in the immediate flanking regions. The *P/R* allele of each strain (*a1* or *a2*) is indicated on the left. Chromosomes inverted relative to their original assembly orientations are marked with asterisks. In panel G, only one representative of *K. europaea* and *K. botswanensis* is shown, and in panel M, only one representative of *K. heveanensis* is included, as the genomes of their mating-type counterparts are not assembled at the chromosome level; the scaffolds containing the *HD* locus are, however, included in the zoomed-in view. Other features are annotated as shown in the key. (PDF)

**S4 Fig. Structure and genomic context of the *P/R* mating-type locus in tetrapolar *Cryptococcus* and *Kwoniella* species.** This supplementary figure spans five pages and contains a total of 15 panels (labeled **A–O**), with three consecutive panels per page. For each group of three panels, the top-left panel shows a synteny view of the full chromosomes, highlighting the chromosomal location of the *P/R* locus; the bottom panel provides a zoomed-in synteny view (~200 kb) that includes the full *P/R* locus; and the top-right panel displays a simplified phylogenetic tree, providing contextual information for the species included in the synteny analyses. For cross-referencing, *P/R*-associated genes in the zoomed-in panel are colored teal if the corresponding ortholog in *Cryptococcus* pathogens is located within the *MAT* locus, shown in a darker shade when positioned in the flanking regions, or colored bright green if the gene is found within the *P/R* locus of some species. The *P/R* allele of each strain (*a1* or *a2*) is indicated on the left. Chromosomes inverted relative to their original assembly orientations are marked with asterisks. In panels A and D, the GC content is depicted as the deviation from the genome average, calculated in 0.5 kb nonoverlapping windows. In panel G, only one representative of *K. europaea* and *K. botswanensis* is shown, and in panel M, only one representative of *K. heveanensis* is included, as the genomes of their mating-type counterparts are not assembled at the chromosome level; the scaffolds containing the *P/R* locus are, however, included in the zoomed-in view. Other features are annotated as shown in the key. (PDF)

**S5 Fig. Predicted rearrangements leading to the extant *P/R* configuration in *Kwoniella* species within clades E/F and clade G.** **(A)** The extant *P/R* locus configuration in *K. mangrovensis* is inferred to derive from a putative ancestral state through three inversion events: (INV1) involving *STE3* and *STE12*, truncating *STE12* in the *a2* allele; (INV2) relocating *DBP10* and *REI1* from the locus edge to its center; and (INV3) moving *NDC80* from the edge to the middle of the locus. **(B)** The extant *P/R* locus configuration in *K. heveanensis*, inferred from a putative ancestral state involving four inversion events: (INV1) involving *STE3* and *STE12*, truncating *STE12* in the a2 allele; (INV1b) inverting only the pheromone gene; (INV2) relocating *DBP10* and *REI1* from the locus edge to its center; (INV3) relocating *BSP3-NDC80* to the middle of the locus; and (INV4) relocating *UBP6* to the center of the locus. INV1 and INV2 are likely ancestral in *Kwoniella*. **(C)** Gene genealogies reveal *trans*-specific polymorphism for *ATG7*, *UBP6*, and *BSP3* within clade G species only. Genes that required manual correction are marked with an asterisk. (PDF)

**S6 Fig. *MAT* locus structure in *Cryptococcus deneoformans*.** Synteny analysis is shown for five strains organized by mating type. Strain JEC20**a** is an F1 progeny of the cross between isolates NIH12α and NIH433**a**, while JEC21α is its congenic partner, generated through 10 rounds of backcrossing to JEC20**a**. Therefore, the *MAT*α allele in JEC21α was inherited from NIH12α, and the *MAT***a** allele in JEC20**a** was inherited from NIH433**a**. The genes *BSP3* and *IKS1* are no longer part of the *MAT* locus in this species. Chromosomes inverted relative to their original assembly orientations are marked with asterisks. (PDF)

**S7 Fig.  *MAT* locus structure in *Cryptococcus gattii* species complex.** This supplementary figure spans 2 pages. **(A)** Synteny analysis with strains organized by mating type and, within each mating type, sorted according to their phylogenetic relationships. The analysis shows that the *MAT***a** configuration is highly variable across species, with each species exhibiting a distinct organization. In contrast, the *MAT*α structure is more conserved across species. **(B)** Synteny analysis with strains grouped by species to compare differences between mating types and define the exact boundaries of the *MAT* locus based on synteny conservation. Note that no *MAT***a** strain of the VGV lineage has been isolated to date. In both panels, chromosomes inverted relative to their original assembly orientations are marked with asterisks.
(PDF)

**S8 Fig.  Phylogenies of *BSP3, PRT1, SPO14,* and *CAP1* within the predicted *MAT* locus of Cryptococcus sp.3.** In pathogenic *Cryptococcus* species, these genes exhibit trans-specific polymorphism, characterized by mating-type-specific clustering, with **a** and α alleles forming distinct clades. An exception is observed in *BSP3* from *C. deneoformans* (marked with asterisks), where mating-type specificity has been lost, representing a derived state. In contrast, protein sequences from *Cryptococcus* sp. 3 (highlighted in bold) do not group with either the **a** or α allele-specific clusters and instead occupy distinct positions outside these groups. Maximum likelihood phylogenies were constructed using protein sequences. For clarity, sequences from other *Cryptococcus* and *Kwoniella* species are collapsed, and well-supported nodes (≥90% SH-aLRT/UFBoot support) are indicated by filled circles. Scale bars represent the number of substitutions per site.
(PDF)

**S9 Fig.  Chromosome number reduction in *Cryptococcus* sp. 3 (CWM60451) involves centromere inactivation via loss of LTR-rich regions. (A)** The karyotype of *K. shandongensis* (with 14 chrs.) served as the reference for reconstructing synteny blocks in pairwise comparisons with representative *Cryptococcus* species of clades A (*C. neoformans* H99), B (*C. amylolentus* CBS6039), C (*Cryptococcus* sp. CMW60451), and D (*Cryptococcus* sp. DSM108351). Each *K. shandongensis* chromosome is assigned a distinct color, which defines the color scheme for homologous synteny blocks in the corresponding chromosomes of other species. Centromeres are labeled "*CEN*" and colored according to their associated reference chromosome. Centromeres inferred to result from intercentromeric recombination are shown with two colors, reflecting ancestry from two distinct *K. shandongensis* centromeres. Inactivated centromeres ("*iCEN*" or "ic") are marked with red arrowheads. **(B)** Linear chromosome plots show synteny conservation among these species and detail the region of centromere inactivation (*iCEN*) in *Cryptococcus* sp. 3. Centromeres are marked with black boxes with white circles and repeat-rich regions are highlighted in pink.
(PDF)

**S10 Fig.  Nuclear genome composition of *Kwoniella mangrovensis* progeny reveals substantial aneuploidy and recombination.** Sequencing read-depth coverage and inheritance patterns of progeny derived from a CBS8507 (*a1b1*) × CBS10435 (*a2b2*) sexual cross. For each progeny, sequencing coverage plots (normalized to the genome-wide average coverage) are color-coded according to each parent's contribution, as shown in the key. Haplotypes blocks inferred from SNP data are overlaid for each chromosome for comparison, reflecting instances of recombination or loss of heterozygosity (LOH). In some strains (MP18, MP50, MP59, and MP54), coverage analysis suggests potential changes in ploidy in a subset of the sequenced cell population. This is indicated by skewed read proportions favoring one parent genome over the other while still retaining the corresponding haplotype, suggestive of genomic instability.
(PDF)

**S11 Fig.  FACS analysis of *Kwoniella mangrovensis* parental strains CBS8507 and CBS10435, and their progeny.** The *C. neoformans* strain JEC21 and XL143 served as haploid (1n), and diploid (2n) controls, respectively.
(PDF)

**S12 Fig. Self-filamentation phenotype of *Kwoniella mangrovensis* progeny.** Parental strains CBS8507 and CBS10435 (grown individually and in co-culture) and their recovered progeny were cultivated on V8, pH 5.0 at room temperature. Self-filamentation was assessed after 2 weeks of incubation. While neither parental strain exhibited self-filamentation in solo culture, their co-culture produced hyphal filaments and basidia.
(TIF)

**S1 Text. GC depletion at the *P/R* locus in tetrapolar *Cryptococcus* is best explained by reduced GC-biased gene conversion.**
(PDF)

**S2 Text. Karyotype reduction in *Cryptococcus* sp. 3 is independent of *P/R–HD* linkage.**
(PDF)

**S1 Appendix.** Genome assembly, genomic features, and information on raw sequencing data generated in this study. **(A)** List of *Cryptococcus* and *Kwoniella* isolates used in this study and summary of genome assembly statistics and other genomic features. **(B)** Genome sequencing, assembly, and polishing approaches. **(C)** NCBI accession numbers of each genome and raw read data generated and used in this study. **(D)** SRA accession numbers of Illumina read data of *C. neoformans MATa* strains used for variant calling and SNP distribution. **(E)** Contig length, centromere coordinates, and telomeric sequences of genome assemblies generated and analyzed in this study.
(XLSX)

**S2 Appendix.** Mating-type (*MAT*) loci regions and genes within *MAT* in *Cryptococcus* and *Kwoniella*. **(A)** Summary of the *MAT* loci regions in bipolar and tetrapolar *Cryptococcus* and *Kwoniella* species. **(B)** List of genes within the *P/R* locus in tetrapolar *Cryptococcus* and *Kwoniella* species and within *MAT* in bipolar species and their essentiality classification. **(C)** *P/R* locus size comparisons between tetrapolar *Cryptococcus* and *Kwoniella* species. **(D)** *P/R* locus size comparisons between tetrapolar *Cryptococcus* species. **(E)** *MAT* locus size comparisons between *MATα* and *MATa* strains of *C. neoformans*. **(F)** *MAT* locus size comparisons between *MATα* and *MATa* strains of the *C. gattii* species complex.
(XLSX)

**S3 Appendix.** Shared and unique mating-type (*MAT*) genes across *Cryptococcus* and *Kwoniella* clades. **(A)** List of genes within *MAT* across *Cryptococcus* and *Kwoniella* species and clades. **(B)** Shared and unique genes within *P/R* locus in *Cryptococcus* and *Kwoniella* clades. **(C)** Shared and unique genes within *MAT* loci (i.e., including *P/R*- and *HD*-associated genes) in *Cryptococcus* and *Kwoniella* clades.
(XLSX)

**S4 Appendix.** GC content and codon composition/usage analysis. **(A)** Genome-wide GC content and deviation from genome mean in clade B and D strains included in Fig 2. **(B)** GC content reduction in the *P/R* locus relative to genome-wide background (clades C and D). **(C)** Neutral comparison of noncoding GC in *P/R* locus vs. chromosome-matched background windows (Test 1). **(D)** Third-position codon usage (AT3 vs. GC3) in *P/R* locus vs. background (Test 2). **(E)** Codon usage and bias metrics in *P/R* vs. matched background genes (Tests 2 and 3). **(F)** Statistical tests for codon usage bias in *P/R* vs. background (Tests 3 and 4). **(G)** Tests for reduced codon adaptation (CAI) after controlling for GC3 and gene length (Test 4).
(XLSX)

**S5 Appendix.** List of primers and PCR-RFLP analysis of *K. mangrovensis* sexual progeny. **(A)** List of primers and conditions used to perform PCR-RFLP. **(B)** List of F1 progeny recovered from CBS8507 × CBS10435 cross and selected for Illumina sequencing after PCR-RFLP analysis. **(C)** List of all F1 progeny recovered from CBS8507 × CBS10435 cross. **(D)** PCR-RFLP results for amplicon MP293-MP294 digested with EcoRV. **(E)** PCR-RFLP results for amplicon MP303-MP304

digested with EcoRV. **(F)** PCR-RFLP results for amplicon MP307-MP308 digested with EcoRV. **(G)** PCR-RFLP results for amplicon MP309-MP310 digested with EcoRV. **(H)** PCR-RFLP results for amplicon MP311-MP312 digested with EcoRI. **(I)** PCR-RFLP results for amplicon MP313-MP314 digested with ClaI.
(XLSX)

## Acknowledgments

We thank Vikas Yadav for assistance with Nanopore data acquisition, Terrance Shea and Christina Cuomo for providing the genomes assemblies of *K. mangrovensis* CBS8886 and CBS10435, as well as *K. heveanensis* BCC8398, and the Broad Institute Genomics Platform for generating Illumina sequencing data for these three strains. We also thank Fred Dietrich for computational resources, Prof Francois Roets for providing permits to collect specimens in South Africa, and Minou Nowrousian for critical reading of this manuscript.

## Author contributions

**Conceptualization:** Marco A. Coelho, Marcia David-Palma, Joseph Heitman.

**Data curation:** Marco A. Coelho, Marcia David-Palma.

**Formal analysis:** Marco A. Coelho.

**Funding acquisition:** Brenda D. Wingfield, Michael J. Wingfield, Joseph Heitman.

**Investigation:** Marco A. Coelho, Marcia David-Palma, Seonju Marincowitz, Janneke Aylward, Nam Q. Pham, Andrey M. Yurkov, Sheng Sun.

**Project administration:** Brenda D. Wingfield, Michael J. Wingfield, Joseph Heitman.

**Resources:** Andrey M. Yurkov, Brenda D. Wingfield, Michael J. Wingfield, Joseph Heitman.

**Software:** Marco A. Coelho.

**Supervision:** Joseph Heitman.

**Visualization:** Marco A. Coelho, Marcia David-Palma.

**Writing – original draft:** Marco A. Coelho, Marcia David-Palma.

**Writing – review & editing:** Marco A. Coelho, Marcia David-Palma, Seonju Marincowitz, Janneke Aylward, Andrey M. Yurkov, Brenda D. Wingfield, Michael J. Wingfield, Sheng Sun, Joseph Heitman.

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
