## [Editor Report · Decision Letter 0]

31 Mar 2025

Dear Joe,

Thank you for submitting your manuscript entitled "Tracing the evolution and genomic dynamics of mating-type loci in Cryptococcus pathogens and closely related species" for consideration as a Research Article by PLOS Biology.

Your manuscript has now been evaluated by the PLOS Biology editorial staff, as well as by an academic editor with relevant expertise, and I'm writing to let you know that we would like to send your submission out for external peer review.

Once your full submission is complete, your paper will undergo a series of checks in preparation for peer review. After your manuscript has passed the checks it will be sent out for review. To provide the metadata for your submission, please Login to Editorial Manager (https://www.editorialmanager.com/pbiology) within two working days, i.e. by Apr 02 2025 11:59PM.

Best wishes,

Roli

Roland Roberts, PhD

Senior Editor

PLOS Biology

rroberts@plos.org

---

## [Decision Letter · Decision Letter 1]

22 May 2025

Dear Joe,

Thank you for your patience while your manuscript "Tracing the evolution and genomic dynamics of mating-type loci in Cryptococcus pathogens and closely related species" went through peer-review at PLOS Biology. Your manuscript has now been evaluated by the PLOS Biology editors, an Academic Editor with relevant expertise, and by four independent reviewers.

You'll see that while reviewer #1 is impressed with the scale of the analysis, s/he feels that it really falls into three separate projects that are not fully tied together by the narrative. S/he suggests restructuring and shortening. There are also some additional suggested analyses and experiments. Reviewer #2 is more positive, but also thinks the paper should be shortened and focussed, perhaps moving material to the supplement. Most of his/her requests are textual or presentational, but they may entail one or two analyses. Reviewer #3 calls this a “tour de force,” though also mentions the length. The requests are textual, with one potential analysis. Reviewer #4 is very positive, but requests some further analyses.

In light of the reviews, which you will find at the end of this email, we are pleased to offer you the opportunity to address the comments from the reviewers in a revision that we anticipate should not take you very long. We will then assess your revised manuscript and your response to the reviewers' comments with our Academic Editor aiming to avoid further rounds of peer-review, although we might need to consult with the reviewers, depending on the nature of the revisions.

**IMPORTANT - SUBMITTING YOUR REVISION**

*Resubmission Checklist*

*Published Peer Review*

*PLOS Data Policy*

*Blot and Gel Data Policy*

Best wishes,

Roli

Roland Roberts, PhD

Senior Editor

PLOS Biology

rroberts@plos.org

REVIEWERS' COMMENTS:

Reviewer #1:

Coelho et al. present an extensive analysis of a large number of genomes from various strains and species of Cryptococcus and Kwoniella fungi, focusing on the mating type (MAT) regions and chromosomes. The study contributes a valuable resource by providing numerous telomere-to-telomere assemblies. While many of the hypotheses explored—such as transitions from tetrapolar to bipolar mating systems, introgression events, and mating type locus rearrangements—have been analyzed before they have not previously been studied in such depth or with such a broad comparative framework. The authors extend the analyses, comparing MAT differences between Kwoniella and Cryptococcus, assess genetic variation within and between these genera. Given that Cryptococcus includes important human pathogens, a detailed understanding of its mating system evolution has clinical as well as evolutionary significance.

The manuscript covers a wide range of topics: introgression of the MATa locus between Cryptococcus lineages, reorganization of the a1 and a2 alleles in Kwoniella, chromosomal fusions and centromere loss, duplications of mating-type pheromones, and gene gains and losses. The strength of the study lies in its comprehensive analysis of MAT loci across these species. However, this breadth also emerges as a weakness: despite the exceptional quality of data and analyses, the overarching narrative of the paper remains unclear. The manuscript attempts to cover too much ground, lacks a unifying hypothesis, and is overly verbose. It reads more like three separate projects—one per genus, plus a comparative component—condensed into a single paper. I will first address this issue, followed by specific comments on the content.

I believe the manuscript would benefit substantially from a restructured organization of topics and a more concise presentation of results.

At present, the flow jumps between different levels of analysis: from overall MAT structure and breeding systems, to P/R in Kwoniella, then to Cryptococcus P/R, back to general MAT structure in Cryptococcus, followed by mating-type chromosome fusions, then back again to Kwoniella chromosomal changes and breeding systems. A more thematic structure—e.g., a main section on "Fusion Events" comparing the two genera, with genus-specific findings as subheadings—would enhance coherence and focus. Figure 10 seems intended to synthesize the major findings, but this integrative perspective is not well reflected throughout the manuscript.

In addition to the structural issues, the writing style contributes to the lack of clarity. The manuscript adopts a storytelling approach that is overly descriptive and often loses sight of the central message. For instance, the section discussing the introgression of the mata locus (lines 409-432) initially suggests that introgression may have occurred, then dismisses this possibility due to structural rearrangements, only to return to the introgression hypothesis based on SNP data. Presenting the conclusion first, followed by supporting evidence, would improve clarity and focus. This issue is also apparent in the introduction: it is lengthy, and only in the final paragraph does it become clear what the paper aims to address. More broadly, the text is excessively wordy, and the authors could significantly reduce its length without sacrificing content or depth.

Comments on content:

Difference between Kwoniella and Cryptococcus:

The authors have two genera that evolved differently, but with convergence in certain aspects. Currently, the manuscript mostly describes these independently, but the comparison of why certain things might have occurred differently are not properly discussed. What are the main ecological differences? What is the population structure of each genus? Why are pheromones expanding in one, but not in the other? It would be interesting if hypotheses are given and tested.

Mating type content:

What is the importance of non-MAT genes in the mating type? The authors mention that in some species there is extension and incorporation of novel genes for cell organelle inheritance for example, however it is unclear why this would happen in general. In isogamous species, the mating types function is mostly limited to regulation of conjugation and a ploidy level physiological switch. Neutral hypotheses are generally not discussed by the authors.

Forces leading to expansion of the regions of reduced recombination are highly debated running from neutral, to beneficial for dosage compensation or heterozygosity (see for example recent papers 10.1016/j.tig.2024.03.005, 10.1126/science.abj1813, 10.1093/evolut/qpae060, 10.1093/evolut/qpad023). The authors tend to imply that gene incorporations are adaptive, however a neutral hypothesis should not be dismissed. For example Line 269: "has expanded to include two additional genes [..] (also present in the MAT locus of Cryptococcus pathogens". This sentence is ambiguous or even misleading. Without giving the synteny of the outgroup, it is unclear if these have been recruited, as is suggested by 'expanded', and 'also present' suggests even that this would be adaptive as it happened twice. I might have missed something, but to me it seems more parsimonious that these genes were lost in the other clades which means they are not adaptive.

Whatever the cause of mating type region expansions, its consequences will be degeneration of genes caught in these regions. To overcome pseudogenization transfer of genes might occur to overcome this degeneration. With the extensive data at hand, including ancestral reconstructions, the authors might be able to analyze this specific aspect of mating type content. For example, as mentioned but not extensively analyzed, fusions of the mating types likely occur for increased sib-sib compatibility but as a consequence will trap many non-mating type related genes in the mating types. Fast reorganization of the mating types might be driven to overcome this degeneration.

In line with this, the very interesting finding that introgression occurred of the mata in Cryptococcus lineages (line 418-) could be associated with degeneration. I suggest to include a reference to Corcoran et al. (10.1101/gr.197244.115), who suggested that introgression of mating types might occur to revive mating type function which due to Muller's ratchet is prone to degenerate. According to their hypothesis introgression might be a means to rejuvenate the genome.

Mating type locus fusions:

About the term 'pseudobipolar': Is it possible to test if the linkage of the two mating types loci is expected based on a neutral scenario, due to the repeated fission and fusion of chromosomes? Or rather that this was likely driven by selection. If this was the only fusion, the chance these two chromosomes merged was 1/(14*13), which is rather small, but if many occur, this could be due to random chance. Even though the distance between the loci is large, it could be that the two loci are linked nevertheless, as recombination can be suppressed over large distances as shown in Neurospora tetrasperma or Microbotryum (10.1038/s41467-017-01317-6, 10.1111/nph.17039).

Breeding system:

It is very exciting to see experimental test of the potential for sexual reproduction. It would be great if there was more known on mating in naturel. Mating types do indicate the potential for haploid selfing or outcrossing, the natural occurrence of outcrossing and inbreeding are not assessed. The authors write: "While genomic analysis of MAT loci content and organization allows for inferences about potential breeding systems (heterothallism vs. homothallism), experimental mating assays remain the most definitive approach for confirming sexual reproduction and viable progeny production." I disagree. The potential for outcrossing, does not indicate the occurrence of outcrossing. Heterothallic species are compatible with a quarter (tetrapolar) or half (bipolar) of their sibs and thus inbreeding can be rife in heterothallics. A better understanding can be obtained by assessing heterozygosity and runs of homozygosity (10.1111/jeb.13780) in aneuploid or diploid natural isolates (when available) or by studying LD decay which is also possible in haploids (10.1073/pnas.1503159112). This will give information not only on whether sexual reproduction is possible, but also actually occurs in nature.

Specific comments:

Figure 1: I really like the figure which gives a great overview of the diversity in the groups! Fig 1A: The groups in 1A are are made somewhat arbitrary: F is a paraphyletic group; the line above Kwoniella bestiolae should be below that species; there should be a group added for K. shivajii. Fig 1C: What is meant by "essential"? The fact that many genes are missing in some species suggests these might not be essential. I assume they are only missing from the mating type region, but still present elsewhere in the genome. It might be informative to add this information too.

Figure S1: Both trees show a species tree. I wonder if the inconsistency between these trees are due to introgression which is obscured in species trees. Gene trees will most likely show different patters, either due to ancestral polymorphism or geneflow. It might be interesting to test the trees per window or chromosome for example. There seems to be more introgression going on (Line 420, and Fig 5, see the region with reduced variants in the middle clade). It would be interesting to know what is going on genome wide.

Line 338: "These findings suggest that the lower GC content of the P/R locus is likely driven by a bias towards AT-rich codons, potentially resulting from mutation accumulation associated with reduced or suppressed recombination in this region." This statement is tautological: GC content is lower because AT content is higher. More importantly, the explanation invoking "mutation accumulation" is somewhat misleading. A more likely cause of reduced GC content in regions with suppressed recombination is a decrease in GC-biased gene conversion (gBGC), which is known to be associated with recombination rate (e.g., 10.1186/s13059-014-0549-1). The authors refer to gBGC in the discussion, but only after introducing a long argument about relaxed selection. While relaxed purifying selection may contribute—especially due to reduced effective population size caused by low recombination (e.g., Hill-Robertson interference) and balancing selection—this should not be the default explanation. Before invoking selection-based mechanisms, the authors should consider testing for patterns consistent with neutral processes. For example, examining whether non-coding regions in the P/R locus show similar GC depletion would help determine whether the GC reduction is due to mutational biases rather than relaxed selection. If the effect is genome-wide or evident in non-coding regions as well, a neutral mechanism like reduced gBGC would be a more parsimonious explanation.

Line 658: Interesting that the pheromones increased in number. The hypothesis that this might occur for mate finding efficiency has been suggested before. Relevant references: 10.1111/j.1420-9101.2012.02539.x, 10.1098/rspb.2008.1146, 10.1111/jeb.12017. I wonder why this is limited to Cryptococcus. If I understand correctly, this mostly occurs in the mata allele, not the matalpha. Could this in some way be associated with single mating type (unisexual) reproduction? Possibly higher pheromone production in mata is selected to overcome the pheromone independent homothallic mating cycle in the matalpha cells (10.1371/journal.pgen.1006772)?

Figure S9 is unclear to me. The colors are not well explained. Are the colors of the chromosomes representing the colors of K. sha? And what about the colors of the centromeres?

Figure 10: This figure could be more powerful by simplification. I think the main story is that 1) mat extensions happened multiple times (numbers 2 and 5) and 2) that mat fusions happened multiple times independently (numbers 3, 4 and 6). By giving too many details, the parallelism is lost as the numbers on the tree that refer to these parallel events differ. The evolutionary forces are likely the same for the expansion and the fusions, so I would number them the same.

Reviewer #2:

The manuscript 'Tracing the evolution and genomic dynamics of mating-type loci in Cryptococcus pathogens and closely related species' by Coelho and colleagues reports on a comprehensive study of the mating type evolution in two genera of basidiomycete fungi. By taking advantage of existing and novel genomic resources, the authors perform comparative genomics to uncover the genome dynamics of mating type loci, resulting in transitions in reproductive strategies, associated with genetic and chromosomal reorganizations at the mating type loci.

This submission extremely comprehensive and collects a large body of exciting and novel data. It is generally well written, and the analyses are generally performed to high standards, using state-of-the-art approaches. The results are presented clearly, figures are of high quality; the overall robustly support the manuscript's conclusion. To accommodate the large body of analyses and results, the manuscript is extremely long with and extensive number of (complex) figures. At the same time, the manuscript's text is also at times challenging to follow for a non-expert in fungal mating and genomics and to pick out what are the key findings. To maybe appeal better to the broad readership of PLoS Biology and to generally improve clarity, it would be beneficial to shorten and focus the manuscript on fewer key observations and either remove others (e.g., coding adaptation, or the in-depth discussion of additional genes associated with mating type loci) from the manuscript or move to S-Material/Text; this is up to the discretion of the authors and the editor.

We have also provided additional detailed comments and suggestions aimed at improving the overall clarity and at addressing few remaining open points of concern:

L98: Dikaryotic hyphal growth is a process already regulated by the canonical HD genes (homeodomain transcription factors) in the MAT locus. HD genes are not 'additional genes', as these are core of the mating type system in all basidiomycetes with a tetrapolar or bipolar mating system. Therefore, attributing dikaryotic growth to the effect of additional genes recruited into fused MAT loci is misleading or imprecise, and this sentence needs to be rephrased.

L157: It would be helpful to clarify whether the sequenced genomes are haploid or dikaryotic. If they are dikaryotic, were the genome assemblies phased? This information is important for interpreting the assembly and analysis results. For example, given that CBS10435, a parental strain used in crosses (Fig. 9C), was found to be diploid (L608), it would be helpful to clarify whether ploidy was confirmed for the other sequenced isolates to ensure consistency in the comparative analyses.

L162: How were the mating types determined to guide the strain selection? Was this done prior to the analyses, based on previous work, or indicated based on observations from this submission. This should be clarified.

L202: In Figure S2A, the individual cultures of C. decagattii strains are shown plated alone, but the yeast morphology is not clearly depicted. For clarity and consistency, it would be helpful to include representative images or insets of the yeast cells for the single-strain cultures, to fully illustrate the yeast-to-hypha transition described in the text; this also applies for Figure S2B. In Figure S2C, for C. depauperatus, it would be helpful to clarify what is shown in the zoomed-in inset for the YPD medium — is this hyphal growth or basidia? Additionally, in the V8 pH 5 condition, the structure appears to be a spore chain, but the text also mentions basidia formation. Basidia are not clearly visible in the current image, it would strengthen the figure to include an additional inset specifically highlighting the basidia.

L227: Few strains were sequenced with Illumina only, resulting in fragmented assemblies. How does this impact the manuscript's results on sizes of mating type loci, organization and distances between genes? It might be useful to indicate genome assembly quality (i.e., which are chromosome-level or not) in Figure 1.

L259 (and other locations, e.g., L320): How were the different mating-associated loci defined? For example, by considering all genes flanked by known mating type genes, or differently? This is obviously relevant for i) determining the size and ii) determine the genes localized within a specific locus. Moreover, there might be genes relevant for mating just outside of the loci, that are currently not considered. The manuscript could be more explicit when it comes to the approach and implication of the mating locus definition. Moreover, in L439 it is mentioned that the two most distal pheromone genes were considered to determine the length. Shouldn't be the minimum distance covering HD genes, STE3, and MF genes be considered, and thus the length should cover both loci.

L275: 'a species' -> "Within Kwoniella species…"?

L284: How were the rearrangement breakpoints determined? By manual evaluation of gene localization, nucleotide alignments, or …? Could you please clarify this?

L339: it is possible that the deviation of AT-rich codons is a signal of introgression rather than an adaptation specific at the MAT locus. Are there additional genomic regions that would show similar deviations (the manuscript reports for enrichment analysis compared to the genome-wide numbers)? Does the phylogeny of genes in MAT loci follow the general species phylogeny? Moreover, in the discussion (L690 and following), the authors later acknowledge that codon usage and expression analyses are needed to better understand P/R locus evolution, in this paragraph they present speculative interpretations about codon usage effects on mating without direct supporting data for the P/R locus itself. I suggest clarifying within this paragraph that these interpretations remain hypothetical and would benefit from future experimental validation.

L483: could the manuscript at this stage provide an intuition how the centromeric positions had been determined? Experimentally using ChIP or computationally using signature transposons/repeats?

L623: while typically not done, it might be useful given the amount of data and length of manuscript to cross-reference between the discussion and figures to enhance readability.

L634/658 (and other): Much of the discussion on mating type systems and their dynamics is focused on basidiomycete yeasts, but little attention is given to rust fungi where recently several manuscripts report about the composition and evolution of mating type loci, with striking analogies to results reported in this manuscript. For example, multiple MF gene copies have been described in these rusts, which should be discussed here. Moreover, the authors suggest that pheromone duplications evolved to strengthen mating partner recognition. However, it would be useful to clarify whether functional validation exists for all the duplicated pheromone genes, specifically, whether they are all actively expressed and contribute to mating compatibility. If not experimentally confirmed, I recommend rephrasing this statement more cautiously. The manuscript also reports multiple STE3 genes in the P/R loci. It would be helpful to clarify whether any of these duplicated genes are known or suspected to be functionally active in mating compatibility, and whether one or several copies are likely to be primarily responsible for partner recognition.

References of MAT loci in rust fungi:

1. Gomez-Zapata, P. A., Tellgren-Roth, C., Samils, B., Stenlid, J., Kaitera, J., Brandstrom Durling, M., ... & Olson, A. (2025). Mating Type Gene Divergence is Associated with Life Cycle Differentiation in Scots Pine Blister Rust. bioRxiv, 2025-05.

2. da Rocha, V. D., Ferreira, E. G. C., Castanho, F. M., Kuwahara, M. K., Godoy, C. V., Meyer, M. C., ... & Marcelino-Guimarães, F. C. (2025). Analysis of the genetic diversity of the soybean rust pathogen Phakopsora pachyrhizi reveals two major evolutionary lineages. Fungal Genetics and Biology, 103990.

3. Sperschneider, J., Chen, J., Anderson, C., Morin, E., Zhang, X., Lewis, D., ... & Dodds, P. (2025). A chromosome-scale genome assembly of the flax rust fungus reveals the two unusually large effector proteins, AvrM3 and AvrN. bioRxiv, 2025-04.

L803: The authors might consider discussing the potential evolutionary link between the transition to bipolar mating systems and the emergence of pathogenicity in Cryptococcus species, as all known pathogenic species are bipolar.

L922: could you provide some details how each round of error correction impacts the errors, as reported by pilon. It is possible that an optimal correction is already achieved after fewer iterations and additional rounds of error correction will actually reduce the quality of the final assembly.

L941: How is the manual annotation performed, i.e., how were the gene models amended? Is this annotation publicly available?

L1743: What are the gCF/sCF values for the other branches? Are these branches well supported?

Figures: it is confusing that the tree in Figure 1a has similar labels as the panels (A-I).

Reviewer #3:

The manuscript by Coelho et al. uses a wide variety of genomic, evolutionary, and mating experiments to reconstruct the evolution of the mating-type loci in the Cryptococcus-Kwoniella clade and, to a certain extent, the evolutionary steps that gave rise to the observed genomic architectures of the mating loci observed in the different species. It is a tour de force that will be widely read and cited by any biologist interested in the evolution of fungal mating systems. The manuscript is long and on the descriptive end of things; that said, it is very clearly written and I had no trouble following the authors' experiments and stream of presentation.

To me, the highlight of the story is the discovery of an independent fusion event of the P/R and HD loci is Cryptococcus species 3 and the independent tetrapolar to bipolar and pseudobipolar transitions in Kwoniella. I share the authors' feeling that the more we sample the diversity of these fungi, the greater the insights into the evolution of these highly dynamic loci will be. By "sampling" the authors seem to focus only sampling additional species, but my sense is that population sampling (especially in instances where populations are structured) will also be of help and may help identify - if they get lucky - potential transitions in the making. Not sure how amenable species in the two genera are for population genomics (especially outside of the pathogens), but it would be worth commenting on this somewhere in the discussion.

I also found it very intriguing that the two switches from tetrapolarity to bipolarity in Cryptococcus took place at or very near the branch that is hard to resolve (the one concerning relationships of clades A, B, and C). Perhaps the mutational events that gave rising to bipolarity in clade A (and perhaps C) also contributed to the reproductive isolation of the three lineages? Are all three topologies roughly equally likely? (i.e., is support for A+B roughly equal to support for A+C as well as to support for B+C?) I know that it is hard to test this hypothesis, but one might expect that such transitions are often accompanied by reproductive isolation of strains with different genomic organizations of the mating locus? Might be worth speculating on this as well.

Minor comments:

- Line 183-184: "forming an early-branching clade": the term "early-branching" is inaccurate - better to phrase this in terms of sister relationships. Maybe "forming a clade that is sister to the rest of the species in the genus Cryptococcus"?

- Line 228: "loci" -> "locus"

- Lines 677-692: relaxation of selection could be tested with dn/ds types of models implemented in software such as PAML and HyPhy.

- Line 819: "most definitive" -> "definitive"

- Line 820 onward: I agree that sampling of new species will be useful. In addition, the authors might want to add a note about the importance of population sampling (especially in species with evidence of structured populations). These genomic structural rearrangements must have come from somewhere and sampling at the population level might enable catching such a rearrangement "in the act" so to speak.

- Figure 10: why invoke two independent transitions to step #5 and not a single one in the branch subtending the two with transitions to step # 5? It's more parsimonious that way

Reviewer #4:

[identifies himself as Aaron A. Vogan]

The manuscript "Tracing the evolution and genomic dynamics of mating-type loci in Cryptococcus pathogens and closely related species" by Coelho et al. is a comprehensive investigation into the genome evolution of species of the genera Kwoniella and Cryptococcus, with a focus on the MAT locus.

This is an excellent manuscript, well written, clear and, encompassing. I have only 1 comment that I feel should be addressed before publishing. This relates to the analysis of introgression of the MAT locus for the VNI strains. The trees are showing the topology of 3 clades to draw conclusion about sister relationships, but this is not possible with only 3 tips. Proper outgroups are required, or more information about how the root was placed. Additionally, it would be more informative to have trees of genes flanking the MAT locus, rather than the whole of chromosome 5 to compare to see how quickly the signal of introgression disappears outside of the MAT locus.

Beyond this I have noticed a very minor issues that can be addressed"

LINE 120: Can one forage for mating partners?

LINE 156: Say how big the previous data set was

LINE 164: One doesn't assemble by long- and short-read sequencing. You generate data form sequencing and then assemble. It should also be clear if this is co-assembly with Illumina or assembly followed by polishing.

Paragraph at 294: Does this suggests a transition from heterothallic to homothallic and back to heterothallic? The transition from homothallic to heterothallic had been considered unlikely and/or unfeasible traditionally, very interesting observation.

FIGURE 6B The C. neo chr_5 and C. sp3 *Chr_11 comparisons show identical chromosome maps on the left and right, but very different synteny. Why?

LINE 466: How do gene phylogenies reflect rearrangements?

LINE 535: I recommend clarifying that it's a reciprocal translocation.

LINE 764: Would recommend not using the name "gypsy" for Ty3.

LINE 891: Sharing stress?

LINE 937: What repeat libraries were used?

---

## [Editor Report · Decision Letter 2]

15 Sep 2025

Dear Joe,

Thank you for the submission of your revised Research Article "The complex evolution and genomic dynamics of mating-type loci in Cryptococcus and Kwoniella" for publication in PLOS Biology. On behalf of my colleagues and the Academic Editor, Sarah Zanders, I'm pleased to say that we can in principle accept your manuscript for publication, provided you address any remaining formatting and reporting issues. These will be detailed in an email you should receive within 2-3 business days from our colleagues in the journal operations team; no action is required from you until then. Please note that we will not be able to formally accept your manuscript and schedule it for publication until you have completed any requested changes.

IMPORTANT: You'll see that we've taken the liberty of changing your Title to something more explicit and (we think) appealing. Do let us know if that's a problem. We'll also need you to add your funders' URLs to your Financial Disclosure statement, so I've asked my colleagues to include this request alongside their own.

Sincerely, 

Roli

Senior Editor

PLOS Biology

rroberts@plos.org